# PPi: Pretraining Brain Signal Model for Patient-independent Seizure Detection

**Zhizhang Yuan*** Zhejiang University zhizhangyuan@zju.edu.cn

**Daoze Zhang*** Zhejiang University zhangdz@zju.edu.cn

**Yang Yang†** Zhejiang University yangya@zju.edu.cn

**Junru Chen** Zhejiang University jrchen_cali@zju.edu.cn

**Yafeng Li** Nuozhu Technology Co., Ltd. yafeng.li@neurox.cn

## Abstract

Automated seizure detection is of great importance to epilepsy diagnosis and treatment. An emerging method used in seizure detection, stereoelectroencephalography (SEEG), can provide detailed and stereoscopic brainwave information. However, modeling SEEG in clinical scenarios will face challenges like huge domain shift between different patients and dramatic pattern evolution among different brain areas. In this study, we propose a **P**retraining-based model for **P**atient-**i**ndependent seizure detection (PPi) to address these challenges. Firstly, we design two novel self-supervised tasks which can extract rich information from abundant SEEG data while preserving the unique characteristics between brain signals recorded from different brain areas. Then two techniques, *channel background subtraction* and *brain region enhancement*, are proposed to effectively tackle the domain shift problem. Extensive experiments show that PPi outperforms the SOTA baselines on two public datasets and a real-world clinical dataset collected by us, which demonstrates the effectiveness and practicability of PPi. Finally, visualization analysis illustrates the rationality of the two domain generalization techniques.

## 1 Introduction

Epilepsy is a chronic disease of brain that affects more than 50 million people worldwide, and a large proportion of patients have drug resistant epilepsy (DRE) which cannot be controlled by medication. Actually 70% of them can live seizure-free only if the *seizure onset zone* (SOZ) can be located and surgically removed [1].

To diagnose epilepsy, the most direct quantitive data to reflect brain function is the electrical activity of the patient's brain. One of the methods to monitor the brain activity is EEG, in which small sensors are attached to the scalp to pick up the electrical signals produced by the brain. Although EEG is widely adopted in brain activity recordings due to its simplicity and relatively low cost, as a non-invasive method, EEG fails to pinpoint the exact seizure focus when the SOZ is located in the deep structure of the brain. In view of the importance of SOZ localization in the treatment of patients with DRE [2], an emerging method called SEEG [3] is applied to locate foci by implanting electrodes in the deep brain. The contacts (also called *channels*) on the electrodes distribute around the suspected lesion across several brain areas, which provide stereoscopic recordings of the brain from both cortical and subcortical structures simultaneously [4].

* Equal contribution.
† Corresponding author.

37th Conference on Neural Information Processing Systems (NeurIPS 2023).

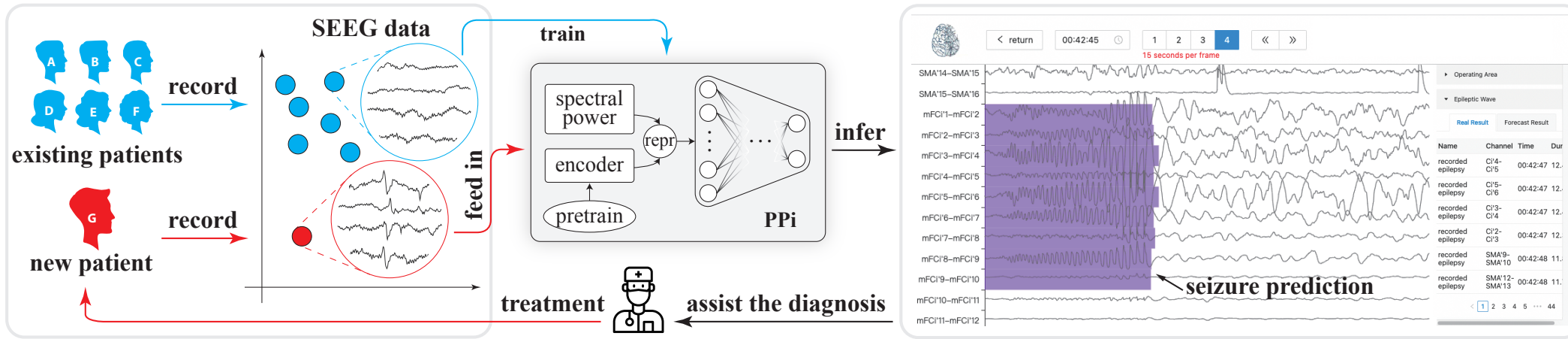

Figure 1: Overview of the automated seizure detection pipeline. Firstly, the model is trained utilizing the SEEG data from existing patients. Then for real-world application, the doctors can upload the SEEG data of new patients and obtain the results of seizure detection predicted by the model, to easily enjoy the diagnostic assistance which is helpful for further treatment.

After collecting the SEEG recordings of patients, the process of epilepsy detection and diagnosis is traditionally treated as a manual task that highly depends on a few experienced neuroscientists, requiring considerable time and human resources [5]. Thus, automating this process could greatly improve the efficiency of clinical seizure detection. As shown in Fig. 1, firstly the model is trained using the labeled data from existing patients, and then the trained model will automatically detect seizures of the SEEG recordings of any new patient. Doctors can view SEEG waveforms and model predictions at the same time (shown on the right side of Fig. 1), and refer to the prediction results for more efficient diagnosis and further treatment. However, existing works for SEEG-based seizure detection mainly focus on the patient-specific setting [6, 7], which can only be trained and directly applied on the same patient due to the substantial differences of SEEG data between patients, resulting in hours even days of training for each new patient. Although a few studies follow the patient-independent setting, these works require manual sampling and denoising by a small number of experienced neurosurgeons, leading to a substantial data bias from real clinical data, which is also time consuming and inapplicable in clinical scenarios.

In fact, designing a model for patient-independent seizure detection on SEEG data under clinical requirements is quite an arduous task due to several unique challenges. Owing to the structural and functional differences of brain neural activities and the variation of invasive electrode numbers and locations (see Fig. 2) caused by individual differences in epileptogenic foci, the seizure patterns of SEEG data are quite different among individuals [8, 9]. Therefore, it is very difficult to perform patient-independent seizure detection on SEEG data. In addition, for epilepsy diagnosis, we have to detect seizures for each monitored brain areas to assist localization of SOZ. Meanwhile, for the fact that the cerebral is composed of multiple brain regions that exert a wide variety of functions [10], the seizure patterns may greatly change across brain areas. Thus, to detect seizures for patients with different lesions, we should preserve the unique characteristics of brain areas. However, this is usually overlooked by existing works and is quite a challenging task under patient-independent setting.

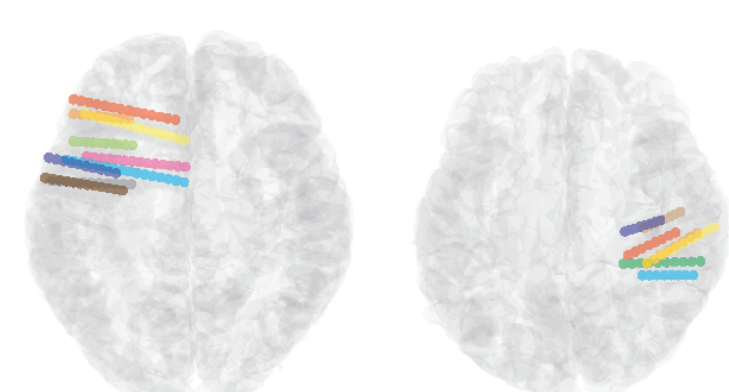

Figure 2: Different SEEG electrode numbers and locations of two patients.

To address the aforementioned challenges above, we propose a novel patient-independent seizure detection model called PPi, which can be successfully applied to clinical SEEG data. PPi adopts a self-supervised learning approach considering significant discrimination of brain areas and contextual coherence of SEEG signals to preserve the patterns of different channels and pretrain on large amount SEEG data. To handle the huge domain shift between patients, we propose channel background subtraction to align the distribution of the same brain region across different patients and brain region enhancement to mitigate the distribution shift brought by different brain regions. In the experimental phase, unlike existing works, we test our model on the real-world clinical data to verify the application value of our method. In summary, our key contributions comprise:

- To the best of our knowledge, we are the first to conduct patient-independent seizure detection on a large-scale real-world clinical SEEG dataset under clinical requirements.

- We propose two novel self-supervised pretraining tasks to preserve the unique patterns of different channels, and two techniques including channel background subtraction and brain region enhancement to handle the domain shift between different patients.
- Extensive experiments show that PPi greatly outperforms the SOTA seizure detection methods (especially on the real-world clinical dataset), demonstrating the application value of our work.

## 2 Related Work

**SEEG-based seizure detection.** SEEG is an emerging method applied in seizure detection, which can localize the SOZ more precisely than those noninvasive recording methods. However, due to the low-quality, large-amount, high-dimensionality characteristics of SEEG data, it is still challenging to develop an automatic approach in SEEG-based seizure detection. Ganti et al. [11] improve seizure detection by temporal Generative Adversarial Networks (TGAN). Chen et al. [6] adopt a graph structure to detect epileptic wave. Xiao et al. [12] propose an SOZ localization method via analyzing the long-term SEEG monitoring for preoperative planning of epilepsy surgery. Although researchers have explored some possible approaches for SEEG-based seizure detection, almost all of these works focus on a patient-specific setting, none of which can be applied in actual clinical scenarios.

**Domain generalization on brain signals.** Our goal is to predict epileptic seizures of SEEG from unseen patients, which can be abstracted as a domain generalization (DG) problem on brain signals. Yang et al. [13] develop a new DG method named ManyDG, that can scale to such many-domain problems for seizure detection task on EEG. Ayodele et al. [14] use transfer component analysis and LSTM to detect epilepsy on EEG data. Jeon et al. [15] propose a mutual information-driven method to conduct subject-invariant and class-relevant deep representation learning of EEG. For these current DG works on brain signals, most of them are conducted on EEG data rather than more informative SEEG. Although Wang et al. [16] study SEEG-based seizure detection on the patient-independent setting, they conduct experiments on datasets which are not only much smaller in size than practical records. The datasets are also manually denoised and sampled to a balanced positive-negative sample ratio which brings about a huge data bias from the real clinical data, indicating that their work is still far from clinical requirements.

**Self-supervised learning on brain signals.** Self-supervised learning is an effective approach when the labeled data is limited. In the field of neural signal (e.g. SEEG, EEG), the label is often hard to obtain. Thus, researchers have developed some SSL methods for this field. Banville et al. [17] utilize relative positioning, temporal shuffling and contrastive predictive coding as the pretext tasks for EEG. Mohsenvand et al. [18] and Kostas et al. [19] model EEG signal using contrastive learning. Cai et al. [20] propose a self-supervised learning framework for brain signals that can be applied to pretrain either SEEG or EEG data. However, these works do not explicitly align the distribution gaps between different domains, which is crucial under the DG setting, especially for data with large domain differences such as SEEG.

## 3 Problem Formulation

Seizure detection on SEEG data can be viewed as a time series classification (TSC) task. The SEEG recording of a patient is a multivariate time series $\mathbf{T} \in \mathbb{R}^{N \times C}$, where $N$ is the length of series, and $C$ is the number of channels. According to the existing works on TSC task [21–23], given an SEEG recording $\boldsymbol{x}_c = (x_1, x_2, \ldots, x_N)$ from channel $c$ of a patient, we divide the contiguous data into small segments to construct data set $\mathbb{S}_c = \{\boldsymbol{s}_{c,0}, \boldsymbol{s}_{c,1}, \ldots, \boldsymbol{s}_{c,K-1}\}$ and the corresponding label set $\mathbb{Y}_c = \{y_{c,0}, y_{c,1}, \ldots, y_{c,K-1}\}$, where $\boldsymbol{s}_{c,k} = \{x_{l\times k+1}, \ldots, x_{l\times(k+1)}\}$ is the $k$-th segment data on channel $c$ from $\mathbf{T}$ ($l$ is the length of each segment, $K = \lfloor N/l \rfloor$ is the total number of segments on channel $c$), and $y_{c,k} \in \{0, 1\}$ is the label of $\boldsymbol{s}_{c,k}$, which indicates whether the segment contains a seizure event ($y_{c,k} = 1$) or not ($y_{c,k} = 0$).

The problem is a DG study in epileptic diagnosis scenario, in which each patient is regarded as a *domain*. Conceptually, DG deals with a challenging setting where one or several different but related domain(s) are given, and the goal is to learn a model that can generalize to an unseen test domain.

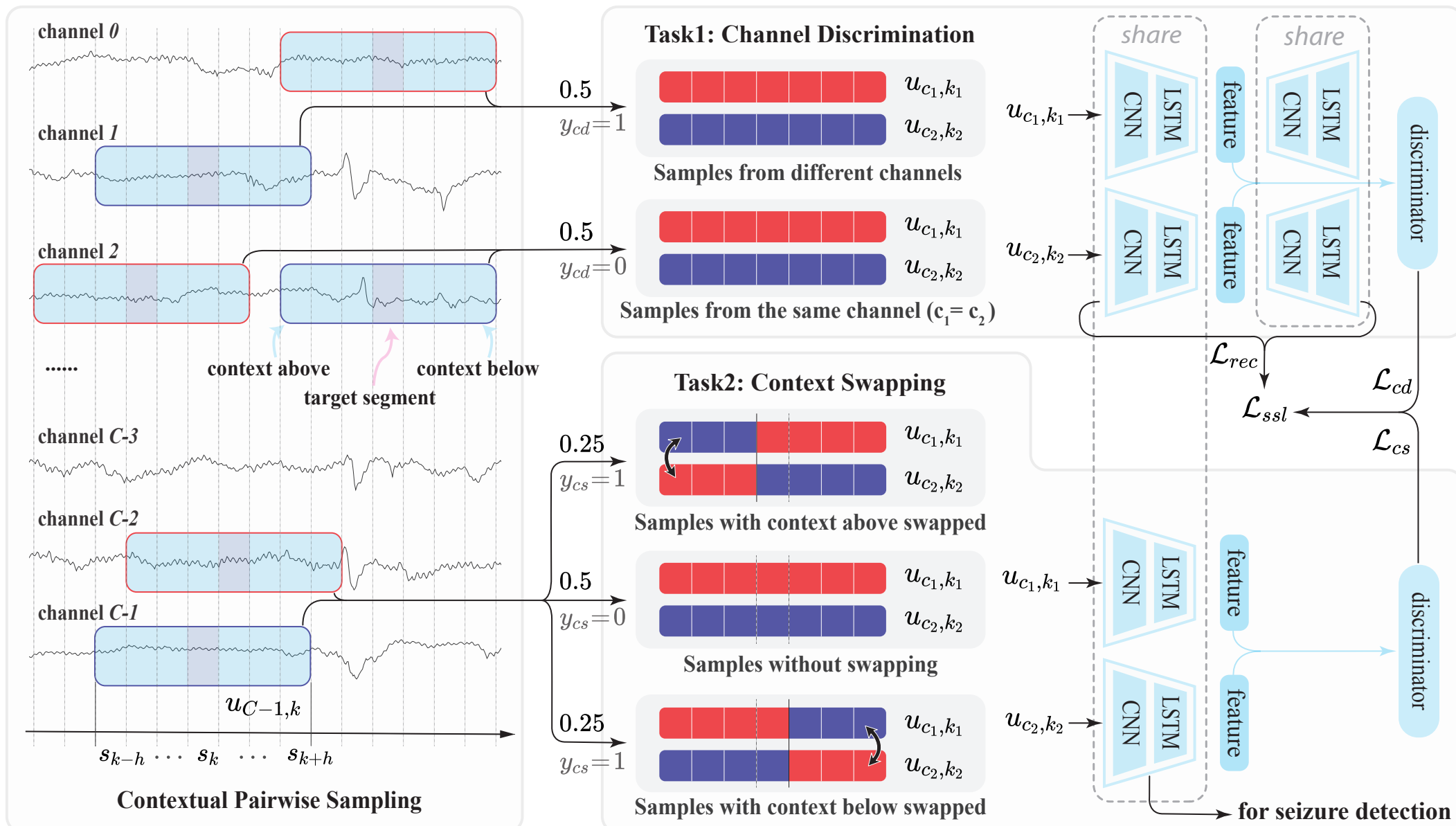

Figure 3: Self-supervised pretraining of PPi. The pretraining is mainly performed by the two self-supervised tasks shown in the figure. In each task, the pairs sampled by contextual pairwise sampling will be properly processed according to the task and a discriminator is employed to discriminate the postive and negative samples (Sec. 4.1.1). The pretrained encoder will be used for the seizure detection (shown in Fig. 4).

Our goal is to utilize the data of labeled patients (*source domains*) to train a model which can be directly adopted to the data unseen patients (*target domains*).

## 4 Methodology

In this section, we introduce the technical details of PPi. As shown in Fig. 3, under the patient-independent setting, we first pretrain an encoder by performing two self-supervised tasks (Sec. 4.1.1) to capture rich information from time domain while preserving unique patterns for each channel, which is consistent with the physiological mechanism of seizures. The pretrained encoder is then applied to the seizure detection task (shown in Fig. 4). To further extract information of SEEG data from a more comprehensive perspective, we also introduce the features from frequency domain by computing the spectral powers in different frequency bands [24] of each segment (Sec. 4.1.2). Finally, based on the learned representations, from time and frequency domains, we adopt channel background subtraction and brain region enhancement techniques to handle the challenge of domain shift (Sec. 4.2).

### 4.1 SEEG Representation Learning

#### 4.1.1 Self-supervised Learning Framework

Self-supervised learning is effective in extracting features for time series [6, 25, 26] and robust to data imbalance [6, 27]. Considering that the SOZ localization is critical for epilepsy diagnosis and treatment, the seizures should be detected on each channel located in different brain areas. However, different brain regions exert a great variety of brain functions [10] and often exhibit different seizure patterns [4]. Therefore, learning representations that preserve the unique patterns of each channel is more consistent with the physiological mechanism of seizures. Thus we design *channel discrimination* and *context swapping* tasks (shown in Fig. 3) considering significant discrimination of channels and contextual coherence of SEEG data, respectively.

**Contextual pairwise sampling.** It is unreasonable to predict whether a short segment is onset or not without contextual data because seizure patterns may vary greatly between different patients and even between different channels of the same patient. To overcome the limitation, we are inspired by the empirical criteria of neurosurgeons in judging seizures that the waveform characteristics of the seizure segment are significantly different from those of nearby normal signals, which is also quantified by Smith [28] as more spikes and sharp waves. Therefore, we introduce the nearby contextual information of a segment in our self-supervised tasks to enhance the discrimination ability of segment representations for detecting seizures. Specifically, the input covers not only the target segment $\boldsymbol{s}_{c,k}$ which will be predicted (defined in Sec. 3), but also the $h$ nearby segments on the left and right, which are called the *context above* and *context below* of $\boldsymbol{s}_{c,k}$ respectively (See Fig. 3). The whole sequence we sample for the target segments $\boldsymbol{s}_{c,k}$ is denoted as $\boldsymbol{u}_{c,k}$. Contextual pairwise sampling is then defined as the operation of randomly sampling two sequences $\boldsymbol{u}^m_{c_1,k_1}, \boldsymbol{u}^m_{c_2,k_2}$ from channel $c_1$ and $c_2$ respectively, where $m$ is the index for the sample set.

**Channel discrimination.** In order to preserve the unique characteristics of each channel, the model should identify how channels are different from each other. Motivated by this, we design a task guiding the model to differentiate whether the given two sequences are from the same channel or not. Specifically, we first sample a sequence pair $\boldsymbol{u}^{m_1}_{c_1,k_1}, \boldsymbol{u}^{m_1}_{c_2,k_2}$ by contextual pairwise sampling, where the two sequences are sampled from the same or different channels with equal probability. Then $\boldsymbol{u}^{m_1}_{c_1,k_1}, \boldsymbol{u}^{m_1}_{c_2,k_2}$ will be encoded to feature vectors $\boldsymbol{h}^{m_1}_{c_1,k_1}, \boldsymbol{h}^{m_1}_{c_2,k_2}$. We next obtain the difference vectors by computing an element-wise absolute difference:

$$\boldsymbol{h}^{m_1}_{cd} = \text{abs}(\boldsymbol{h}^{m_1}_{c_1,k_1} - \boldsymbol{h}^{m_1}_{c_2,k_2}). \tag{1}$$

After that, we utilize a discriminator to predict whether the sampled sequence pair comes from the same channel or not. We apply the binary cross-entropy, denoted as $\mathcal{L}_{cd}$, as the loss function of the channel discrimination task. Meanwhile, to avoid representation collapse and exploit the information from time domain, we apply a decoder to reconstruct the original sequences, denoted as $\hat{\boldsymbol{u}}^{m_1}_{c_1,k_1}, \hat{\boldsymbol{u}}^{m_1}_{c_2,k_2}$. Based on $M_1$ sequence pairs sampled in the channel discrimination task, the objective function for the reconstruction task is defined as:

$$\mathcal{L}_{rec} = \sum_{m_1=1}^{M_1} (\|\boldsymbol{u}^{m_1}_{c_1,k_1} - \hat{\boldsymbol{u}}^{m_1}_{c_1,k_1}\|^2 + \|\boldsymbol{u}^{m_1}_{c_2,k_2} - \hat{\boldsymbol{u}}^{m_1}_{c_2,k_2}\|^2). \tag{2}$$

**Context swapping.** Given that the model leverages contextual information to detect seizures, we also need to enhance the coherence semantic uniqueness of SEEG data. We then propose a task leading the model to identify whether the context has been replaced by that of other channels. First, we also adopt contextual pairwise sampling to sample two sequences from different channels. For the sampled sequence pair, we perform the swapping operation according to the following rules: (1) swap their context above with a probability of $0.25$; (2) swap their context below with a probability of $0.25$; (3) do not swap, otherwise. We do not perform reconstruction task here by reason of possibly corrupted sequences. After the processed data is fed into the encoder, we concatenate the encoded representations of the sequence pair to obtain the joint vectors. Finally, an MLP-based discriminator is utilized to discriminate whether the joint vectors are from swapped sequence pair or not. The binary cross-entropy, denoted as $\mathcal{L}_{cs}$, is also employed as the objective function of the context swapping.

Putting the objective functions all together, the overall self-supervised model will be jointly trained according to the objective function given by $\mathcal{L}_{ssl} = \mathcal{L}_{rec} + \mathcal{L}_{cd} + \mathcal{L}_{cs}$.

#### 4.1.2 Frequency Domain Features

The proposed self-supervised learning tasks have a good capability of extracting features from time domain. To exploit SEEG data from a more comprehensive view, we adopt PSD (power spectral density), which has the ability to track the transient changes before and during seizure [29], to extract features from frequency domain. The PSD of a signal describes the distribution of the signal's total average power over frequency. Specifically, we first split the frequency domain into several bands according to the standard description for rhythmic activity [24]: (1) $\theta$ (4-8Hz), (2) $\alpha$ (8-13Hz), (3) $\beta$ (13-30Hz), (4) $\gamma 1$ (30-50Hz), (5) $\gamma 2$ (50-70Hz), (6) $\gamma 3$ (70-90Hz), (7) $\gamma 4$ (90-110Hz), (8) $\gamma 5$ (110-128Hz). The absolute spectral power of a signal in a frequency band is then computed as the

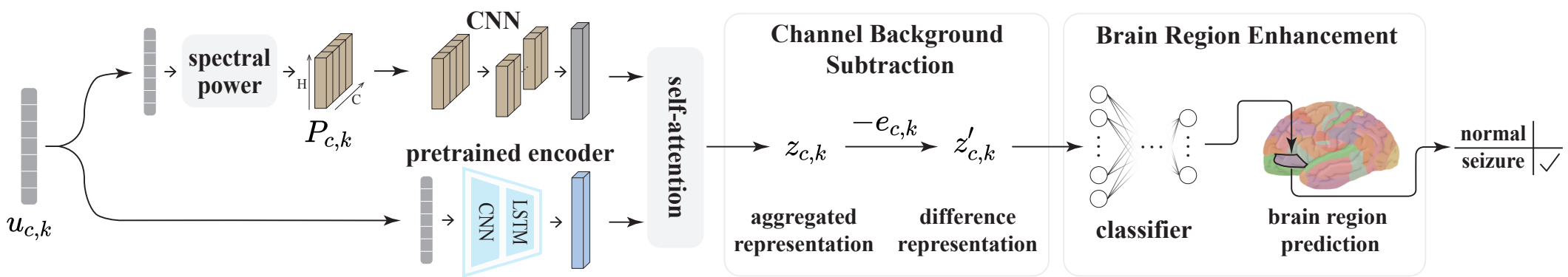

Figure 4: Overall architecture of PPi. The representations encoded from the pretrained encoder and spectral powers (Sec. 4.1.2) are aggregated by a self-attention strategy to obtain the aggregated representation $\boldsymbol{z}_{c,k}$. $\boldsymbol{z}_{c,k}$ is then be processed by channel background subtraction and brain region enhancement to obtain the difference representation $\boldsymbol{z}'_{c,k}$ for seizure detection (Sec. 4.2).

logarithm of the sum of PSD coefficients within that frequency band. Mathematically, for segment $\boldsymbol{s}_{c,k}$, the absolute spectral power in the $i$-th frequency band is computed as:

$$P^i_{c,k} = \log \sum_{\omega \in \text{band}(i)} PSD_{\boldsymbol{s}_{c,k}}(\omega), \quad i \in \{1, 2, \dots, 8\}. \tag{3}$$

In order to capture underlying correlations between different frequency bands and local patterns of contextual segments, the spectral powers will be encoded by a CNN-based encoder to obtain the representations from frequency domain of SEEG signals.

After obtaining the time domain representations from self-supervised learning tasks and frequency domain representations from spectral powers, a self-attention strategy is used to adaptively aggregate them to obtain the segment representations $\boldsymbol{z}_{c,k}$.

### 4.2 Patient-independent Seizure Detection

In order to solve the domain shift problem, we propose an assumption from a higher-order perspective that the differential distribution of normal and seizure segments of channels located in the same brain region would be similar in different patients. The assumption is reasonable because of the similarity in structure and function of the same brain region among patients, and the case study in Sec. 5.6 further supports it. Therefore, based on the assumption, we design two techniques called channel background subtraction and brain region enhancement, which fully consider the characteristics of SEEG, to align the data distribution between patients.

**Algorithm 1** Background Representation Calculation

**for** $c = 0, 1, \dots, C-1$ **do**
  $\boldsymbol{e}_{c,0} \leftarrow \boldsymbol{z}_{c,0}$
  $n_c \leftarrow 1$
  $\hat{y}_{c,0} \leftarrow 0$ [1]
  **for** $k = 1, 2, \dots, K-1$ **do**
    $\boldsymbol{e}_{c,k} \leftarrow \boldsymbol{e}_{c,k-1}$
    **if** $\hat{y}_{c,k-1} < 0.5$ **then**
      $\boldsymbol{e}_{c,k} \leftarrow \frac{\boldsymbol{e}_{c,k} \times n_c + \boldsymbol{z}_{c,k-1}}{n_c + 1}$
      $n_c \leftarrow n_c + 1$
    **end if**
    $\boldsymbol{z}'_{c,k} \leftarrow \boldsymbol{z}_{c,k} - \boldsymbol{e}_{c,k}$
    Predict the seizure probability $\hat{y}_{c,k}$ using $\boldsymbol{z}'_{c,k}$
  **end for**
**end for**

**Channel background subtraction.** Channel background subtraction enables the model to exclusively focus on the differences between seizure and normal segments of each channel, which aligns the distribution of channels within the same brain region across patients. In the field of neuroscience, Staba and Worrell [30] prove that the background brain activity, which refers to the brain activity when individuals are at rest or during sleep, contains important information about brain function and dysfunction in epilepsy. In our scenario, given the patient's limited mobility while monitored by SEEG, background is regarded as the SEEG signal in the absence of epileptic seizures. Therefore, we calculate the background representation $\boldsymbol{e}_{c,k}$ for the segment representation $\boldsymbol{z}_{c,k}$ as the average of all the segment representations *currently* (before index $k$) predicted to be normal in channel $c$. We show the details of the iterative updating method in Algo. 1. In channel background subtraction, $\boldsymbol{z}_{c,k}$ will subtract $\boldsymbol{e}_{c,k}$ to obtain the difference representation $\boldsymbol{z}'_{c,k}$. The operation aligns the distribution of channels located in the same brain region between patients, which is supported by the case study in Sec. 5.6.

[1] Since the patient must be in normal state at the start of the SEEG recording, it is reasonable to set $\hat{y}_{c,0}$ to 0.

**Brain region enhancement.** Channel background subtraction has aligned channel representations with similar latent differential distributions. However, with prior information of brain regions, explicit supervised learning can further constrain the differential distribution of channels in the same brain region between different patients, thus enhancing the generalization of PPi. Therefore, we extend the classification task from binary to multi-class, so that the classifier not only predicts whether the seizure occurs, but also determines which brain region the seizure locates in[2]. Specifically, taking $\boldsymbol{z}'_{c,k}$ as input, the classifier outputs a tuple $(i_{c,k}, y_{c,k})$, where $i_{c,k}$ is the index of the brain region that $s_{c,k}$ locates in and $y_{c,k} \in \{0, 1\}$ is the prediction of whether $s_{c,k}$ is seizure or not. For the implementation, we construct integer multi-class labels by converting the tuple $(i_{c,k}, y_{c,k})$ to an integer $y'_{c,k}$: $y'_{c,k} = i_{c,k} * 2 + y_{c,k}$. Therefore, the parity of $y'_{c,k}$ reflects whether the prediction result corresponds to a normal state or a seizure. We use a cross-entropy loss as the objective function of brain region enhancement.

# 5 Experiments

## 5.1 Dataset

**Public datasets.** The public datasets used in our paper, MAYO and FNUSA [32], are collected from two institutions: the Mayo Clinic (Rochester, Minnesota, United States of America) and St. Anne's University Hospital (Brno, Czech Republic), respectively. The MAYO dataset has 18 patients in total, including 56730 normal samples and 15227 seizure samples, respectively. The FNUSA dataset has 13 patients in total, including 94560 normal samples and 52470 seizure samples, respectively. For each dataset, we first remove the power line noise and down sample the dataset to 500Hz. Then we divide the patients into 6 groups without overlapping (details in App. B). We randomly choose 5 groups as the source domains (4 of which are used for training and 1 for validation) and the remaining group serves as the target domain. The experiments are repeated on all groups to test the average performance.

**Clinical dataset.** The clinical SEEG dataset we collect is from a first-class hospital. For the patients, 4 to 10 invasive electrodes with 52 to 126 channels are implanted in the brain to obtain 1000Hz SEEG signals. Since the clinical data are recorded with a high frequency on multiple channels, the dataset has more than 738 hours of recording and contains 123 patient files with an average size of 7.1 GB each. As for the annotation, professional neurosurgeons participate in seizure labeling. The positive sample ratio of a single patient in the dataset is around 0.004 on average, which is extremely imbalanced. We remove the power line noise and down sample the dataset to 250Hz. For the 7 patients in the clinical dataset, we split the patients into training, validation and test set with 5, 1 and 1 patients, respectively. We also repeat the experiments on all patients to obtain an overall results.

## 5.2 Experimental Setup

To evaluate the performance of the models, we conduct adequate experiments on two public datasets (MAYO and FNUSA) and the real-world clinical dataset. All experiments run on a Linux system with 2 CPUs (AMD EPYC 7H12 64-Core Processor) and 4 GPUs (NVIDIA GeForce RTX 3090). Our code is available at `https://github.com/yzz673/PPi_public`.

**Evaluation Metrics.** To comprehensively evaluate the experimental results, we use precision, recall, F1 and F2 as evaluation metrics. Here the F-score can be calculated by $F_\beta = \frac{(1+\beta^2)\times precision \times recall}{\beta^2 \times precision + recall}$. Usually, F2 is adopted in critical applications that value information retrieval more than accuracy (i.e., accepting a relatively large number of false positives but virtually guaranteeing that all the true positives are found). In our medical scenario, F2 is more valued than F1, since ignoring any seizure is costly in diagnosis.

**Baselines.** We compare our model with some DG methods for brain signals (SICR [15], SEEG-Net [16]). Also, we compare with other DG algorithms designed for more general fields

[2]In our study, brain regions are divided according to a medical standard template called *automated anatomical labeling* (AAL) [31], which is a digital atlas of the human brain. AAL defines 116 different regions in total, of which 90 are in the cerebrum and 26 are in the cerebellum.

(CDANN [33], CORAL [34], GroupDRO [35], MLDG [36], MMD [37], MTL [38], SANDMask [39], SD [40], SelfReg [41], TRM [42], VREx [43], IB-ERM [44], IB-IRM [44] ) and adopt two different feature extractors (TCN [45], MiniRocket [46]) to evaluate their performance based on different kinds of features. Furthermore, we select some self-supervised learning approaches for brain signals (BENDR [19]) and time series (Franceschi et al. [47]). More details of the baselines are shown in App. A.

Table 1: Average performance of patient-independent seizure detection tasks on two public datasets and our clinical dataset. The **v** indicates the first in a column, v indicates the second, and *v indicates the third. The performance with standard deviation is given in App. C.

| | Model \ Dataset | MAYO | | | | FNUSA | | | | Clinical | | | |
|---|---|---|---|---|---|---|---|---|---|---|---|---|---|
| | | Pre. | Rec. | F1 | F2 | Pre. | Rec. | F1 | F2 | Pre. | Rec. | F1 | F2 |
| TCN | CDANN | 16.13 | *68.14 | 24.74 | 37.84 | 32.71 | 69.62 | 43.09 | 54.88 | 1.42 | 44.20 | 2.72 | 6.08 |
| | CORAL | 17.01 | 64.08 | 25.73 | 38.30 | 33.87 | 65.46 | 43.63 | 53.84 | 1.59 | 47.24 | 3.03 | 6.64 |
| | GroupDRO | 17.26 | 64.25 | 26.02 | 38.68 | 33.24 | 62.14 | 41.90 | 51.27 | 1.53 | 47.13 | 2.91 | 6.35 |
| | MLDG | 18.88 | 61.38 | 27.96 | *40.39 | 32.81 | 56.81 | 40.41 | 48.24 | 28.85 | 6.68 | 2.55 | 2.58 |
| | MMD | 15.79 | 67.41 | 24.45 | 37.68 | 32.87 | *69.38 | 43.51 | 55.30 | 1.90 | 46.84 | 3.26 | 6.59 |
| | MTL | 16.13 | 51.87 | 22.56 | 31.74 | 32.70 | 53.10 | 38.48 | 44.78 | 1.43 | 9.19 | 2.30 | 3.73 |
| | SANDMask | 16.47 | 68.66 | 25.00 | 38.15 | 32.16 | 59.46 | 39.62 | 48.59 | 1.34 | *47.22 | 2.60 | 5.95 |
| | SD | 15.29 | 60.70 | 23.10 | 34.67 | 34.30 | 65.54 | 43.77 | 53.88 | 1.52 | 46.46 | 2.91 | 6.38 |
| | SelfReg | 9.42 | 58.65 | 15.50 | 26.29 | 25.35 | 66.75 | 35.87 | 48.79 | 0.92 | 44.48 | 1.76 | 3.94 |
| | TRM | 16.94 | 59.60 | 24.47 | 35.67 | 34.32 | 62.26 | 43.39 | 51.93 | 1.53 | 47.34 | 2.91 | 6.35 |
| | VREx | 16.74 | 60.41 | 25.06 | 36.87 | 33.17 | 61.75 | 41.84 | 51.11 | 10.98 | 42.17 | 2.78 | 5.33 |
| | IB-ERM | 17.27 | 64.29 | 26.04 | 38.71 | 34.34 | 62.94 | 42.93 | 52.22 | 1.53 | 47.34 | 2.91 | 6.35 |
| | IB-IRM | 16.75 | 60.45 | 25.08 | 36.90 | 34.26 | 63.06 | 43.10 | 52.44 | 1.52 | 45.65 | 2.89 | 6.31 |
| MiniRocket | CDANN | 16.22 | 15.44 | 15.14 | 15.20 | 48.73 | 36.97 | 38.26 | 37.02 | 2.09 | 45.27 | 3.98 | 8.68 |
| | CORAL | 47.61 | 25.30 | 27.57 | 25.46 | 68.46 | 46.55 | 52.50 | 48.36 | 1.68 | 42.54 | 3.20 | 7.08 |
| | GroupDRO | 41.99 | 34.02 | 35.98 | 34.49 | 69.79 | 48.74 | 55.31 | 50.83 | 1.37 | 46.95 | 2.64 | 6.03 |
| | MLDG | 10.32 | 50.04 | 15.35 | 24.70 | 34.67 | 63.56 | 37.88 | 46.75 | 0.52 | 15.58 | 1.01 | 2.29 |
| | MMD | 22.34 | 21.82 | 19.58 | 20.33 | 69.33 | 46.63 | 50.04 | 47.06 | 5.10 | 39.97 | 3.95 | 7.55 |
| | MTL | 21.67 | 46.11 | 27.72 | 35.03 | 56.85 | 59.71 | *56.93 | 58.28 | 12.57 | 45.79 | 4.02 | 5.15 |
| | SANDMask | 4.32 | 33.33 | 7.40 | 13.14 | 12.56 | 33.33 | 18.14 | 24.88 | 1.49 | 44.87 | 2.59 | 4.73 |
| | SD | 37.57 | 30.90 | 32.46 | 31.10 | 68.74 | 50.59 | 53.26 | 50.92 | 6.40 | 39.73 | 9.43 | 15.29 |
| | SelfReg | 31.09 | 14.46 | 18.02 | 15.65 | 61.18 | 33.26 | 39.29 | 34.99 | 16.33 | 42.06 | *14.51 | 16.61 |
| | TRM | 35.97 | 40.34 | 35.51 | 37.15 | 66.32 | 53.63 | 53.50 | 52.68 | 5.00 | 33.31 | 7.81 | 12.51 |
| | VREx | 38.63 | 33.92 | 33.37 | 32.88 | 65.23 | 55.20 | 54.34 | 53.91 | 7.47 | 44.50 | 11.37 | *17.11 |
| | IB-ERM | 35.02 | 43.92 | 36.79 | 39.62 | 66.34 | 53.41 | 53.17 | 52.40 | 4.94 | 43.01 | 8.12 | 13.79 |
| | IB-IRM | 36.27 | 43.22 | *37.36 | 39.57 | 64.67 | 54.02 | 53.69 | 53.00 | 8.56 | 44.63 | 11.18 | 15.06 |
| | BENDR | 23.26 | 45.02 | 25.90 | 30.23 | 40.45 | 37.22 | 34.42 | 34.01 | 2.48 | 28.99 | 3.58 | 5.79 |
| | Franceschi et al. | 34.21 | 40.98 | 33.94 | 35.15 | 43.28 | 50.56 | 44.03 | 48.97 | 2.62 | 44.74 | 4.26 | 9.65 |
| | SICR | 10.18 | 4.56 | 6.10 | 5.06 | 23.51 | 7.16 | 9.79 | 8.01 | *25.34 | 29.22 | 9.19 | 9.80 |
| | SEEG-Net | *45.41 | 45.62 | 43.54 | 44.22 | *69.39 | 53.75 | 60.02 | *55.99 | 20.06 | 32.81 | 20.82 | 22.92 |
| | PPi | **49.85** | **69.67** | **54.35** | **61.07** | **71.73** | **70.81** | **70.61** | **70.55** | **29.76** | **47.59** | **30.92** | **35.51** |

### 5.3 Experimental Result

Tab. 1 summarizes the main results of our model and baselines on the two public datasets and the clinical dataset. Overall, our model outperforms all baselines on every metric on the three datasets, which demonstrates the excellent performance of PPi in the patient-independent seizure detection on SEEG data.

On the public datasets, PPi improves the performance[3] by *38.10%* and *24.98%* over the best-performing baseline model in terms of F2-score respectively, showing that PPi has stronger generalization ability in seizure detection than other baselines. On our clinical dataset, PPi improves the performance by *54.93%* on F2-score. Compared with other baselines that treat all channels equally, our designed self-supervised tasks can preserve more unique characteristics of each channel. These tasks help our model to learn more informative representations from SEEG data, leading to a much better performance on channel-level seizure detection. For SEEG-Net [16] which also focuses on patient-independent seizure detection on SEEG, it can maintain a more balanced precision and recall than other baselines, and achieves almost the second/third highest F1-score in all datasets. The possible reason for its good performance is that SEEG-Net also considers the contextual information of SEEG data like our model. However, PPi still outperforms SEEG-Net by a large margin, as PPi

[3]Here we calculate the relative improvement.

not only preserve unique characteristics of each channel, but also introduces the extra information of brain regions to enhance the generalization ability. In particular, we achieve better improvement on clinical dataset (*54.93%* on F2-score) compared with the public datasets (*38.10%* and *24.98%* on F2-score) mainly because the self-supervised pretraining on unlabeled data reduces reliance on labels, which makes PPi more adaptive to extremely imbalanced dataset than other supervised methods.

### 5.4 Ablation Study

To evaluate the effectiveness of each component in our model, we first conduct ablation experiments on four model variants to verify the effectiveness of the proposed self-supervised framework, including: (1) PPi-$SSL_1$: PPi without the channel discrimination task; (2) PPi-$SSL_2$: PPi without the context swapping task; (3) PPi-$SSL$: PPi without all self-supervised tasks; (4) PPi-reconstruction: PPi without reconstruction loss. Then we use PPi-power: PPi without features of spectral powers, to demonstrate the benefits of introducing PSD based features. For the aggregation strategy, we replace the self-attention with mean pooling (denoted as PPi-self attention). Another two experiments are conducted to verify the significant generalization ability of our two techniques respectively, which are denoted as PPi-background and PPi-brain region. Due to the requirement of brain region labeling in brain region enhancement, the lack of such labels makes brain region enhancement inapplicable to the two public datasets.

Table 2: Results of ablation experiments on two public datasets and the clinical dataset.

| Model \ Dataset | MAYO | | | | FNUSA | | | | Clinical | | | |
|---|---|---|---|---|---|---|---|---|---|---|---|---|
| | Pre. | Rec. | F1 | F2 | Pre. | Rec. | F1 | F2 | Pre. | Rec. | F1 | F2 |
| PPi-$SSL_1$ | 43.69 | 48.01 | 31.22 | 35.37 | 58.46 | 59.39 | 56.05 | 57.27 | 20.68 | 44.23 | 19.13 | 24.63 |
| PPi-$SSL_2$ | 40.69 | 38.32 | 34.72 | 35.74 | 47.69 | 50.53 | 46.79 | 48.42 | 20.75 | 42.82 | 18.72 | 23.73 |
| PPi-$SSL$ | 35.65 | 22.89 | 25.92 | 23.77 | 69.00 | 44.29 | 53.04 | 47.29 | 7.53 | 30.75 | 10.94 | 16.08 |
| PPi-reconstruction | 44.01 | 50.29 | 34.15 | 36.89 | 60.78 | 62.31 | 59.48 | 60.02 | 22.04 | 44.84 | 21.23 | 25.01 |
| PPi-power | 48.60 | 31.43 | 29.01 | 28.31 | 62.89 | 54.48 | 56.77 | 55.13 | 28.13 | 30.48 | 19.75 | 20.87 |
| PPi-self attention | 48.82 | 60.20 | 51.15 | 55.41 | 65.14 | 66.91 | 62.17 | 63.98 | 28.67 | 46.98 | 29.56 | 32.59 |
| PPi-background | 46.80 | 31.31 | 31.08 | 30.05 | 57.83 | 50.20 | 51.68 | 50.28 | 20.42 | 35.40 | 21.83 | 26.74 |
| PPi-brain region | - | - | - | - | - | - | - | - | 18.01 | 46.86 | 21.53 | 28.33 |
| PPi | **49.85** | **69.67** | **54.35** | **61.07** | **71.73** | **70.81** | **70.61** | **70.55** | **29.76** | **47.59** | **30.92** | **35.51** |

The comparison results of the ablation experiments on all the three datasets are presented in Tab. 2. It shows that PPi beats other model variants on all metrics, proving the contribution of each component in our model. The performances of the variants that remove the self-supervised tasks drop greatly, which shows that our designed self-supervised tasks are capable of extracting informative representations from large amount imbalanced SEEG data. PPi-background and PPi-brain region cannot achieve performances that are competitive with the full model, validating the strong generalization ability of PPi to handle the huge domain shift across different patients. The improvement is empowered by the channel background subtraction and brain region enhancement techniques, which will be explored further in Sec. 5.6.

### 5.5 Hyperparameter Analysis

The contexts half length $h$ (i.e. the length of context above or context below) is an important hyperparameter in PPi. Thus we evaluate the performance on the clinical dataset under different contexts half length (shown in Fig. 5). The results exhibits an increase of performance as $h$ becomes larger, which illustrate that introducing the contexts of the target segment to increase the receptive field allows the model to compare the target segment with its nearby waveforms, resulting in better performance. However, the increase of contexts half length is also accompanied by a higher computational overhead, necessitating consideration of trade-offs when selecting the value of $h$.

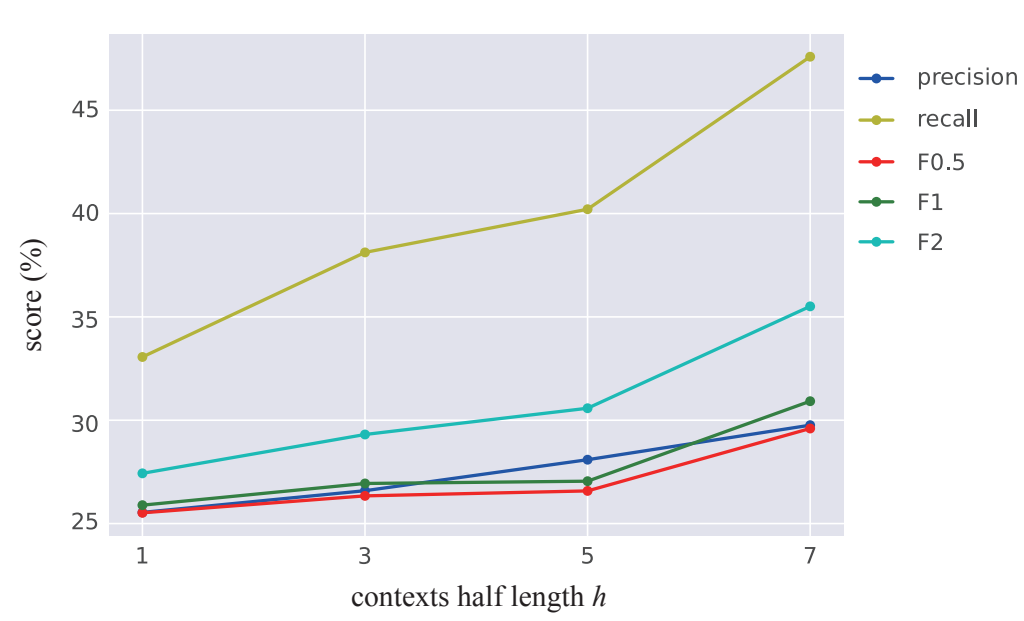

Figure 5: Performance on clinical dataset under different contexts half length $h$.

### 5.6 Case Study

**Visual analysis of channel background subtraction.** To handle domain shift, PPi adopts channel background subtraction which has the ability to align the distribution of the same brain region from different patients. In the visual analysis, we visualize such ability by comparing the aggregated representation $\boldsymbol{z}_{c,k}$ (before subtraction) and the difference representation $\boldsymbol{z}'_{c,k}$ (after subtraction) using t-SNE [48] plots. Specifically, we select two channels which locate in the same brain region from two different patients (denoted as $p_i$, $p_j$) (results for more patients are shown in App. D.1). The t-SNE plots of their aggregated representations and difference representations are shown in Fig 6. The aggregated representations of $p_i$ and $p_j$ obviously exhibit different distributions. After channel background subtraction, the distributions of the samples from $p_i$ and $p_j$ are very close, which illustrates that channel background subtraction successfully aligns the distribution space of the channels located in the same brain region between patients.

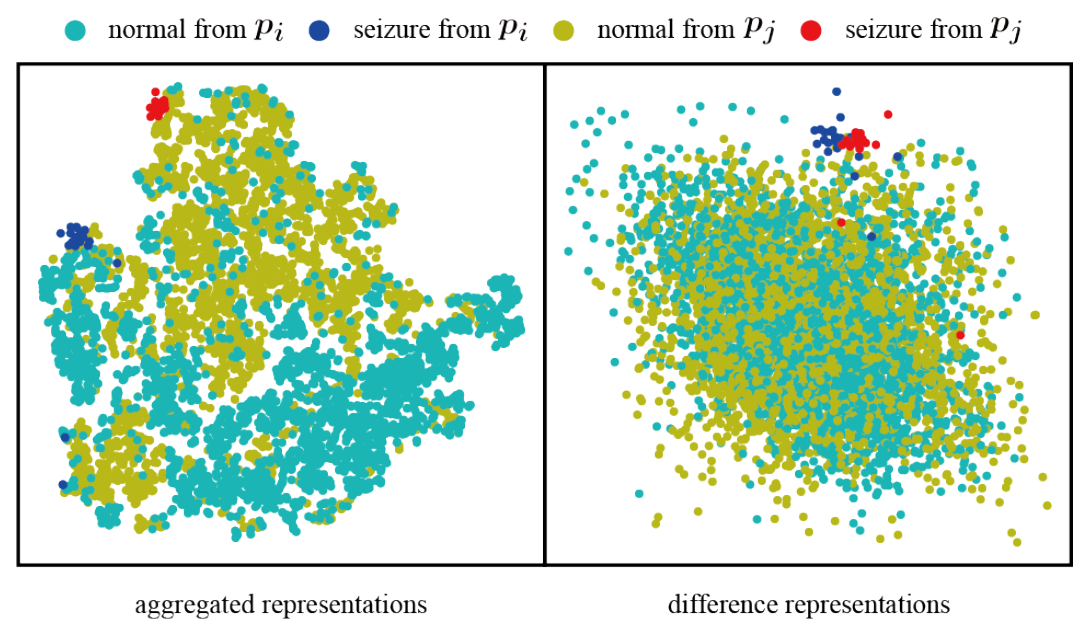

Figure 6: t-SNE plots of aggregated representations (left) and difference representations (right) of two channels from the same brain region of patients $p_i$ and $p_j$.

**Confusion matrix of brain region enhancement.** In order to demonstrate the effectiveness of brain region enhancement, we calculate the confusion matrix of the multi-classification. Fig. 7 shows the confution matrix from one of the patients (the confusion matrix of all the patients are shown in App. D.2), in which the vertical axis represents the multi-class label and the horizontal axis represents the multi-class prediction results. In the confusion matrix, most samples are distributed on the main diagonal, which reflects the good performance of the multi-classification task, illustrating the effectiveness of brain region enhancement.

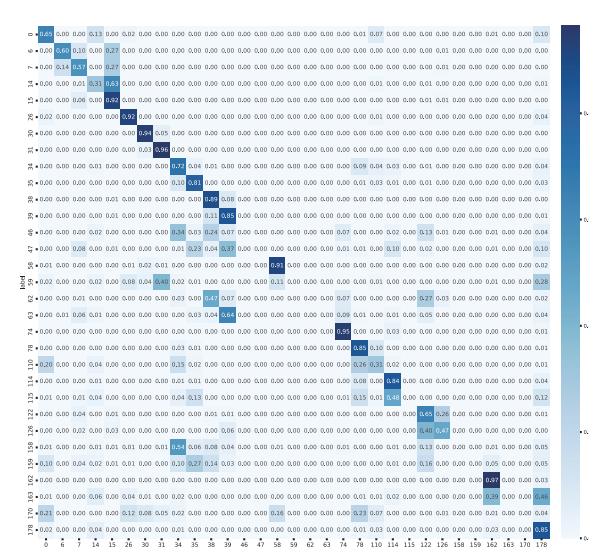

Figure 7: Confusion matrix of brain region enhancement.

## 6 Conclusion

In this paper, we propose PPi to conduct patient-independent seizure detection on SEEG in the clinical scenario. To detect seizures more accurately, PPi adopts a self-supervised pretraining strategy to extract information from SEEG signals while preserving the unique characteristics of each channel. Furthermore, we propose channel background subtraction and brain region enhancement to improve the generalization ability of PPi. Extensive experiments demonstrate the superior performance of PPi in the patient-independent seizure detection on two public and a clinical SEEG datasets (Sec. 5.3). The case study (Sec. 5.6) further illustrate the effectiveness of our proposed two techniques to reduce the huge domain shift between different patients.

**Limitations and future works.** Compared with EEG, SEEG is an emerging technique and the related research is limited. Although our work outperforms other methods by a large margin on the clinical dataset, in applications, the predicted results of PPi are mainly serve as a reference to assist doctors to achieve more efficient clinical diagnosis and treatment, rather than completely replace doctors in seizure detection. For the application, we have reached a cooperation with a first-class hospital and a related institution that is responsible for the development of an application software. In the future, our model will be integrated into the software to assist doctors in seizure diagnosis.

## Acknowledgement

This work is supported by NSFC (No.62176233), the National Key Research and Development Project of China (No.2018AAA0101900) and the Fundamental Research Funds for the Central Universities.

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

# A Details of Baselines

Firstly, we compare our model to some cross-subject methods on brain signals:

- SICR [15]: a framework that learns class-relevant and subject-invariant feature representations, which shows a promising performance in non-invasive brain-computer interface.
- SEEG-Net [16]: a model that can address the problems of sample imbalance, cross-subject domain shift, and poor interpretability and realizes high-sensitivity SEEG pathological activity detection. The source code of SEEG-Net is not released, so we implement it by ourselves to conduct the experiments.

Secondly, to further compare our model to some DG algorithms in more general areas, experiments were conducted with methods as follows:

- CDANN [33]: an end-to-end conditional invariant deep DG approach by leveraging deep neural networks for domain-invariant representation learning.
- CORAL [34]: an unsupervised domain adaptation method that aligns the second-order statistics of the source and target distributions with a linear transformation.
- GroupDRO [35]: a model coupling group DRO models with increased regularization, where DRO allows to learn models that instead minimize the worst-case training loss over a set of groups.
- MLDG [36]: a model agnostic training procedure for DG, which simulates train/test domain shift during training by synthesizing virtual testing domains within each mini-batch.
- MMD [37]: an adversarial autoencoder framework to learn a generalized latent feature representation across domains.
- MTL [38]: a representative framework for DG, which augments the original feature space with the marginal distribution of feature vectors.
- SANDMask [39]: a masking strategy, which determines a continuous weight based on the agreement of gradients, in order to control the amount of update in each step of optimization under the notion of Out-of-Distribution (OOD) Generalization.
- SD [40]: a regularization method aimed at decoupling feature learning dynamics, improving accuracy and robustness in cases hindered by gradient starvation.
- SelfReg [41]: a regularization method for DG based on contrastive learning, self-supervised contrastive regularization.
- TRM [42]: a robust estimation criterion that is specifically geared towards optimizing transfer to new environments.
- VREx [43]: a penalty on the variance of training risks as a simpler variant based on a form of robust optimization over a perturbation set of extrapolated domains.
- IB-ERM [44]: a DG method that improve generalization via minimizes the empirical risk over multiple domains.
- IB-IRM [44]: a DG method that improve generalization via minimizes the invariant risk over multiple domains.

Furthermore, we choose some self-supervised learning approaches on brain signals or general time series as our baselines:

- BENDR [19]: A self-supervised training model that learn compressed representations to model completely novel raw EEG sequences recorded with differing hardware, and different subjects performing different tasks.
- Franceschi et al [47]. This work combines an encoder based on causal dilated convolutions with a novel triplet loss employing time-based negative sampling, obtaining general-purpose representations for variable length and multivariate time series.

# B Details of Datasets

## B.1 Public Datasets

The group dividing strategy and detailed statistics of the two public datasets are shown in Tab. 3. The datasets are publicly available to use under CC0 license and might be downloaded from `https://springernature.figshare.com/collections/Multicenter_intracranial_EEG_dataset_for_classification_of_graphoelements_and_artifactual_signals/4681208`.

Table 3: Group dividing strategy and detailed statistics of two public datasets.

| Group id \ Dataset | MAYO | | | | Group id \ Dataset | FNUSA | | | |
|---|---|---|---|---|---|---|---|---|---|
| | Patient id | Artifacts | Seizure | Normal | | Patient id | Artifacts | Seizure | Normal |
| Group1 | 0 | 2318 | 0 | 330 | Group1 | 1 | 0 | 1912 | 0 |
| | 18 | 1700 | 0 | 3126 | | 5 | 5059 | 1527 | 5452 |
| | 21 | 58 | 3432 | 0 | | | | | |
| | Total | 4076 | 3432 | 3456 | | Total | 5059 | 3439 | 5452 |
| Group2 | 1 | 0 | 883 | 8653 | Group2 | 2 | 2892 | 1657 | 7809 |
| | 9 | 740 | 0 | 0 | | 9 | 0 | 6750 | 0 |
| | 19 | 5613 | 0 | 0 | | | | | |
| | Total | 6353 | 883 | 8653 | | Total | 2892 | 8407 | 7809 |
| Group3 | 2 | 466 | 1923 | 399 | Group3 | 3 | 12 | 8076 | 0 |
| | 5 | 1002 | 0 | 6583 | | 4 | 8463 | 0 | 0 |
| | 16 | 3699 | 0 | 177 | | 12 | 1343 | 7710 | 38217 |
| | Total | 5167 | 1923 | 7159 | | Total | 9818 | 15786 | 38217 |
| Group4 | 3 | 4636 | 0 | 2057 | Group4 | 6 | 0 | 1554 | 962 |
| | 4 | 2063 | 0 | 790 | | 7 | 5416 | 7738 | 2689 |
| | 23 | 761 | 2747 | 644 | | | | | |
| | Total | 7460 | 2747 | 3491 | | Total | 5416 | 9292 | 3651 |
| Group5 | 6 | 12873 | 0 | 0 | Group5 | 8 | 18 | 1896 | 20860 |
| | 7 | 0 | 0 | 25951 | | 10 | 5786 | 4260 | 1545 |
| | 8 | 0 | 2816 | 0 | | | | | |
| | Total | 12873 | 2816 | 25951 | | Total | 5804 | 6156 | 22405 |
| Group6 | 14 | 0 | 3426 | 498 | Group6 | 11 | 3339 | 4072 | 2890 |
| | 17 | 4096 | 0 | 6098 | | 13 | 181 | 5318 | 14136 |
| | 20 | 1278 | 0 | 1424 | | | | | |
| | Total | 5374 | 3426 | 8020 | | Total | 3520 | 9390 | 17026 |

## B.2 Clinical Dataset

The detailed information of the clinical dataset is shown in Tab. 4 and the sample rate of all the patients is 1000Hz.

Table 4: Details information of the clinical dataset.

| Patient id | Time (hours) | #Electrodes | #Channels | Positive sample ratio | #Timestamps |
|---|---|---|---|---|---|
| 0 | 121.4 | 10 | 126 | 0.0028 | 180632184 |
| 1 | 34.7 | 4 | 52 | 0.0020 | 21642853 |
| 2 | 167.7 | 10 | 126 | 0.0011 | 241488147 |
| 3 | 73.7 | 8 | 116 | 0.0077 | 102649651 |
| 4 | 161.3 | 8 | 112 | 0.0037 | 195457035 |
| 5 | 54.3 | 7 | 93 | 0.0016 | 43684998 |
| 6 | 125.2 | 5 | 59 | 0.0167 | 70605073 |

# C Experimental Results with Standard Deviation

The average performance with standard deviation of MAYO, FNUSA and clinical dataset are shown in Tab. 5, Tab. 6 and Tab. 7, respectively.

# D Full Results of Case Study

## D.1 Visual Analysis of Channel Background Subtraction

In order to show that the visual analysis in the case study( 5.6) is consistent across the patients in our clinical dataset, we plot six groups of t-SNE figures (shown in Fig. 8). In each group, the samples are from the same brain region of two patients, and the patients involved in these six examples include all the patients in the clinical dataset. For each group, the left and right are plotted with the representations before and after channel background subtraction. Overall, in each group of experiments, channel background subtraction shows a similar effect, indicating that the effect of channel background subtraction is consistent across patients.

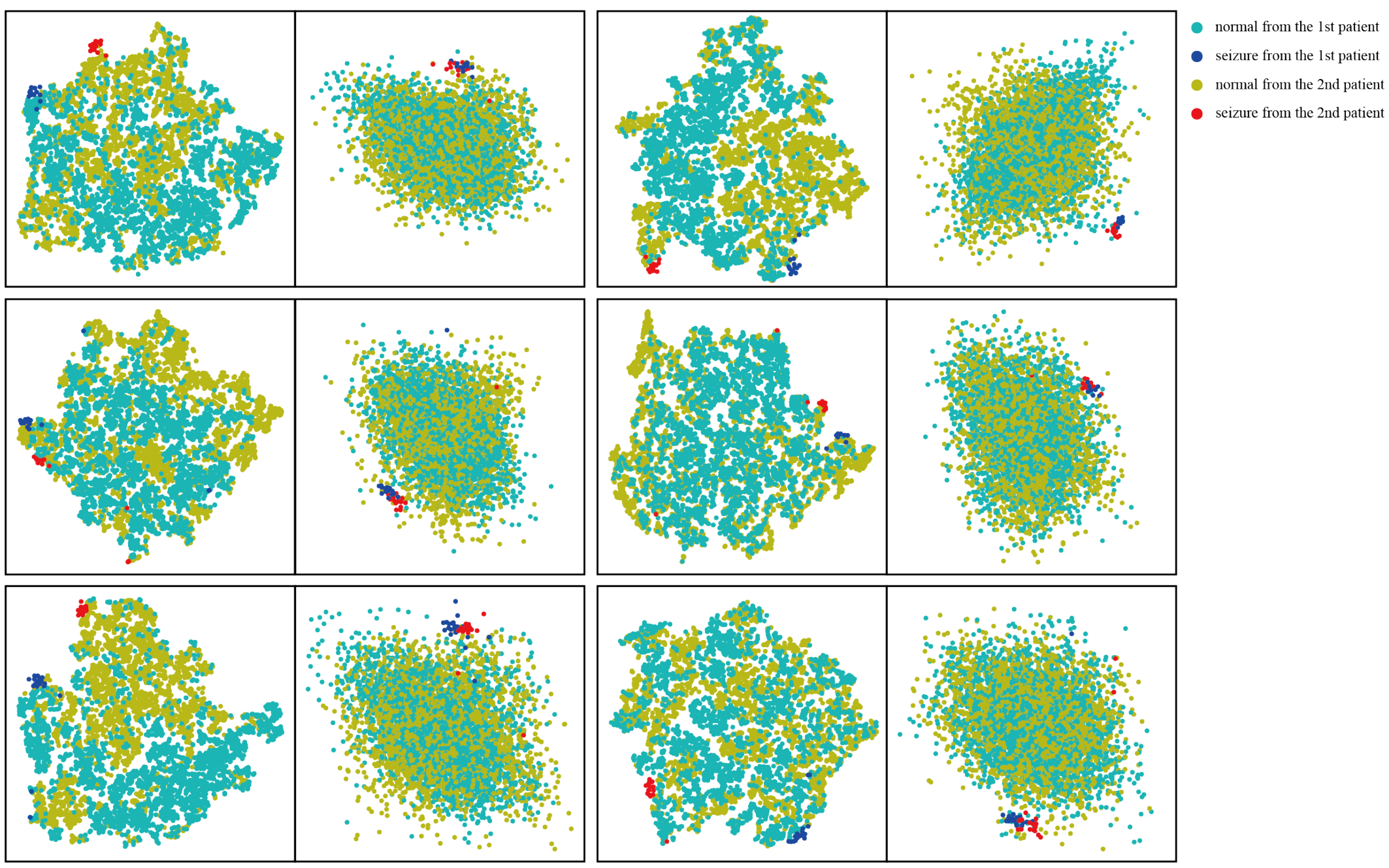

Figure 8: Six visualization examples as a supplement and improvement to the visual analysis of channel background subtraction.

## D.2 Confusion Matrix of Brain Region Enhancement

The confusion matrix of multi-classification on all the patients are shown in Fig. 9. Note that the total number of classes in the confusion matrix is different for different patients. This is because the number and positions of electrodes implanted in each patient are different when the SEEG signal is collected, resulting in different brain regions involved by the SEEG data of different patients, which is also one of the reasons for domain shift.

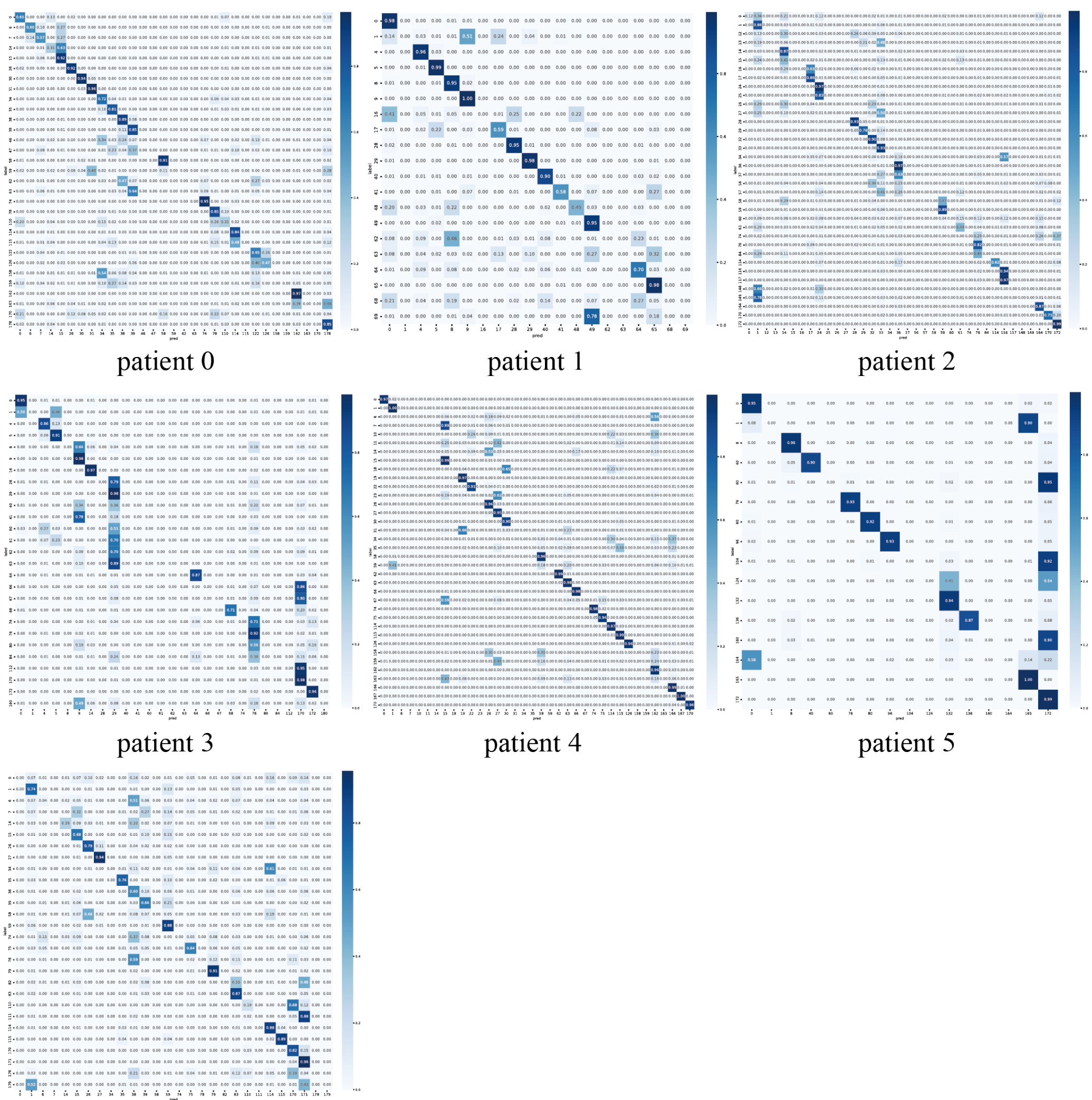

Figure 9: Confusion matrix of brain region enhancement from the multi-classification results of all patients in the clinical dataset.

Table 5: Average performance with standard deviation of patient-independent seizure detection tasks on MAYO. The **v** indicates the first in a column, v indicates the second, and *v indicates the third.

| Model \ Dataset | | MAYO | | | |
|---|---|---|---|---|---|
| | | Pre. | Rec. | F1 | F2 |
| TCN | CDANN | 16.13 ± 5.80 | 68.14± 4.17 | 24.74 ± 7.67 | 37.84 ± 8.74 |
| | CORAL | 17.01 ± 5.14 | 64.08 ± 3.97 | 25.73 ± 6.63 | 38.30 ± 7.31 |
| | GroupDRO | 17.26 ± 5.10 | 64.25 ± 2.32 | 26.02 ± 6.43 | 38.68 ± 6.82 |
| | MLDG | 18.88 ± 4.22 | 61.38 ± 4.65 | 27.96 ± 4.92 | 40.39± 4.78 |
| | MMD | 15.79 ± 4.77 | 67.41 ± 7.21 | 24.45 ± 6.28 | 37.68 ± 7.34 |
| | MTL | 16.13 ± 4.72 | 51.87 ± 11.96 | 22.56 ± 5.11 | 31.74 ± 5.72 |
| | SANDMask | 16.47 ± 4.84 | *68.66± 12.66 | 25.00 ± 6.62 | 38.15 ± 8.64 |
| | SD | 15.29 ± 5.08 | 60.70 ± 3.01 | 23.10 ± 6.38 | 34.67 ± 6.69 |
| | SelfReg | 9.42 ± 3.80 | 58.65 ± 14.87 | 15.50 ± 5.55 | 26.29 ± 7.81 |
| | TRM | 16.94 ± 4.82 | 59.60 ± 5.18 | 24.47 ± 5.30 | 35.67 ± 5.36 |
| | VREx | 16.74 ± 5.18 | 60.41 ± 3.59 | 25.06 ± 6.57 | 36.87 ± 7.01 |
| | IB_ERM | 17.27 ± 5.10 | 64.29 ± 2.30 | 26.04 ± 6.44 | 38.71 ± 6.82 |
| | IB_IRM | 16.75 ± 5.19 | 60.45 ± 3.60 | 25.08 ± 6.58 | 36.90 ± 7.03 |
| MiniRocket | CDANN | 16.22 ± 1.94 | 15.44 ± 8.29 | 15.14 ± 5.49 | 15.20 ± 7.20 |
| | CORAL | *47.61± 14.14 | 25.30 ± 9.35 | 27.57 ± 10.35 | 25.46 ± 9.55 |
| | GroupDRO | 41.99 ± 15.59 | 34.02 ± 13.71 | 35.98 ± 14.05 | 34.49 ± 13.70 |
| | MLDG | 10.32 ± 3.16 | 50.04 ± 12.47 | 15.35 ± 5.36 | 24.70 ± 7.85 |
| | MMD | 22.34 ± 10.38 | 21.82 ± 8.95 | 19.58 ± 6.87 | 20.33 ± 7.49 |
| | MTL | 21.67 ± 7.15 | 46.11 ± 12.08 | 27.72 ± 7.38 | 35.03 ± 7.90 |
| | SANDMask | 4.32 ± 5.24 | 33.33 ± 0.00 | 7.40 ± 8.25 | 13.14 ± 11.81 |
| | SD | 37.57 ± 7.52 | 30.90 ± 6.21 | 32.46 ± 6.00 | 31.10 ± 5.76 |
| | SelfReg | 31.09 ± 2.33 | 14.46 ± 6.28 | 18.02 ± 6.85 | 15.65 ± 6.53 |
| | TRM | 35.97 ± 7.11 | 40.34 ± 8.15 | 35.51 ± 5.86 | 37.15 ± 5.92 |
| | VREx | 38.63 ± 7.60 | 33.92 ± 8.63 | 33.37 ± 6.66 | 32.88 ± 7.15 |
| | IB_ERM | 35.02 ± 7.18 | 43.92 ± 7.49 | 36.79 ± 6.25 | 39.62 ± 5.65 |
| | IB_IRM | 36.27 ± 7.56 | 43.22 ± 6.55 | 37.36± 5.92 | 39.57 ± 4.71 |
| BENDR | | 23.26 ± 4.23 | 45.02 ± 8.14 | 25.90 ± 5.47 | 30.23 ± 6.01 |
| Franceschi et al. | | 34.21 ± 4.96 | 40.98 ± 7.29 | 33.94 ± 6.33 | 35.15 ± 6.72 |
| SICR | | 10.18 ± 6.38 | 4.56 ± 2.72 | 6.10 ± 3.75 | 5.06 ± 3.04 |
| SEEG-Net | | 45.41± 9.96 | 45.62 ± 9.56 | *43.54± 8.84 | *44.22± 8.98 |
| PPi | | **49.85**± 6.93 | **69.67**± 2.82 | **54.35**± 4.72 | **61.07**± 4.69 |

Table 6: Average performance with standard deviation of patient-independent seizure detection tasks on FNUSA. The **v** indicates the first in a column, v indicates the second, and *v indicates the third.

| Model \ Dataset | | FNUSA | | | |
|---|---|---|---|---|---|
| | | Pre. | Rec. | F1 | F2 |
| TCN | CDANN | 32.71 ± 6.42 | 69.62± 2.77 | 43.09 ± 5.61 | 54.88 ± 3.58 |
| | CORAL | 33.87 ± 6.25 | 65.46 ± 3.33 | 43.63 ± 5.78 | 53.84 ± 4.44 |
| | GroupDRO | 33.24 ± 6.43 | 62.14 ± 2.99 | 41.90 ± 5.24 | 51.27 ± 3.47 |
| | MLDG | 32.81 ± 6.61 | 56.81 ± 2.41 | 40.41 ± 5.43 | 48.24 ± 3.71 |
| | MMD | 32.87 ± 6.48 | *69.38± 1.55 | 43.51 ± 5.98 | 55.30 ± 4.28 |
| | MTL | 32.70 ± 6.44 | 53.10 ± 11.03 | 38.48 ± 6.63 | 44.78 ± 7.54 |
| | SANDMask | 32.16 ± 6.31 | 59.46 ± 11.63 | 39.62 ± 5.99 | 48.59 ± 7.87 |
| | SD | 34.30 ± 6.11 | 65.54 ± 1.85 | 43.77 ± 5.08 | 53.88 ± 3.16 |
| | SelfReg | 25.35 ± 7.72 | 66.75 ± 16.60 | 35.87 ± 9.95 | 48.79 ± 12.41 |
| | TRM | 34.32 ± 6.17 | 62.26 ± 4.01 | 43.39 ± 5.10 | 51.93 ± 3.76 |
| | VREx | 33.17 ± 6.37 | 61.75 ± 2.59 | 41.84 ± 5.25 | 51.11 ± 3.42 |
| | IB_ERM | 34.34 ± 6.15 | 62.94 ± 3.88 | 42.93 ± 4.93 | 52.22 ± 3.55 |
| | IB_IRM | 34.26 ± 6.14 | 63.06 ± 2.77 | 43.10 ± 5.05 | 52.44 ± 3.33 |
| MiniRocket | CDANN | 48.73 ± 12.53 | 36.97 ± 9.74 | 38.26 ± 6.56 | 37.02 ± 8.43 |
| | CORAL | 68.46 ± 7.01 | 46.55 ± 11.81 | 52.50 ± 8.63 | 48.36 ± 10.57 |
| | GroupDRO | 69.79± 5.37 | 48.74 ± 11.65 | 55.31 ± 8.35 | 50.83 ± 10.35 |
| | MLDG | 34.67 ± 10.16 | 63.56 ± 16.37 | 37.88 ± 5.75 | 46.75 ± 7.83 |
| | MMD | 69.33 ± 6.50 | 46.63 ± 15.10 | 50.04 ± 8.88 | 47.06 ± 12.62 |
| | MTL | 56.85 ± 5.25 | 59.71 ± 10.71 | *56.93± 6.94 | 58.28± 9.00 |
| | SANDMask | 12.56 ± 4.47 | 33.33 ± 4.02 | 18.14 ± 4.72 | 24.88 ± 3.59 |
| | SD | 68.74 ± 11.63 | 50.59 ± 12.30 | 53.26 ± 6.88 | 50.92 ± 10.09 |
| | SelfReg | 61.18 ± 17.04 | 33.26 ± 11.38 | 39.29 ± 10.66 | 34.99 ± 10.86 |
| | TRM | 66.32 ± 12.14 | 53.63 ± 12.07 | 53.50 ± 5.78 | 52.68 ± 9.35 |
| | VREx | 65.23 ± 9.83 | 55.20 ± 9.33 | 54.34 ± 5.36 | 53.91 ± 7.58 |
| | IB_ERM | 66.34 ± 12.17 | 53.41 ± 12.38 | 53.17 ± 5.82 | 52.40 ± 9.57 |
| | IB_IRM | 64.67 ± 11.99 | 54.02 ± 11.29 | 53.69 ± 5.41 | 53.00 ± 8.55 |
| BENDR | | 40.45 ± 7.02 | 37.22 ± 6.44 | 34.42 ± 6.23 | 34.01 ± 6.10 |
| Franceschi et al. | | 43.28 ± 5.90 | 50.56 ± 8.39 | 44.03 ± 7.11 | 48.97 ± 7.68 |
| SICR | | 23.51 ± 14.12 | 7.16 ± 5.66 | 9.79 ± 7.87 | 8.01 ± 6.37 |
| SEEG-Net | | *69.39± 9.23 | 53.75 ± 7.62 | 60.02± 8.05 | *55.99± 7.73 |
| PPi | | **71.73**± 4.06 | **70.81**± 2.14 | **70.61**± 2.82 | **70.55**± 2.28 |

Table 7: Average performance with standard deviation of patient-independent seizure detection tasks on clinical dataset. The **v** indicates the first in a column, v indicates the second, and *v indicates the third.

| Model \ Dataset | | Clinical | | | |
|---|---|---|---|---|---|
| | | Pre. | Rec. | F1 | F2 |
| TCN | CDANN | 1.42 ± 0.51 | 44.20 ± 13.85 | 2.72 ± 0.98 | 6.08 ± 2.21 |
| | CORAL | 1.59 ± 0.64 | 47.24 ± 2.12 | 3.03 ± 1.20 | 6.64 ± 2.53 |
| | GroupDRO | 1.53 ± 0.60 | 47.13 ± 1.40 | 2.91 ± 1.12 | 6.35 ± 2.35 |
| | MLDG | 28.85± 9.34 | 6.68 ± 3.92 | 2.55 ± 0.58 | 2.58 ± 0.89 |
| | MMD | 1.90 ± 0.70 | 46.84 ± 8.69 | 3.26 ± 1.16 | 6.59 ± 2.52 |
| | MTL | 1.43 ± 0.56 | 9.19 ± 0.59 | 2.30 ± 0.85 | 3.73 ± 1.18 |
| | SANDMask | 1.34 ± 0.34 | *47.22± 4.77 | 2.60 ± 0.65 | 5.95 ± 1.40 |
| | SD | 1.52 ± 0.60 | 46.46 ± 3.56 | 2.91 ± 1.13 | 6.38 ± 2.38 |
| | SelfReg | 0.92 ± 0.62 | 44.48 ± 5.04 | 1.76 ± 1.18 | 3.94 ± 2.52 |
| | TRM | 1.53 ± 0.60 | 47.34± 1.42 | 2.91 ± 1.12 | 6.35 ± 2.35 |
| | VREx | 10.98 ± 5.60 | 42.17 ± 8.25 | 2.78 ± 0.67 | 5.33 ± 1.25 |
| | IB_ERM | 1.53 ± 0.60 | 47.34± 1.42 | 2.91 ± 1.12 | 6.35 ± 2.35 |
| | IB_IRM | 1.52 ± 0.60 | 45.65 ± 2.69 | 2.89 ± 1.13 | 6.31 ± 2.38 |
| MiniRocket | CDANN | 2.09 ± 0.83 | 45.27 ± 15.67 | 3.98 ± 1.58 | 8.68 ± 3.47 |
| | CORAL | 1.68 ± 0.66 | 42.54 ± 19.84 | 3.20 ± 1.30 | 7.08 ± 3.04 |
| | GroupDRO | 1.37 ± 0.62 | 46.95 ± 7.15 | 2.64 ± 1.19 | 6.03 ± 2.63 |
| | MLDG | 0.52 ± 0.91 | 15.58 ± 21.61 | 1.01 ± 1.75 | 2.29 ± 3.92 |
| | MMD | 5.10 ± 5.16 | 39.97 ± 19.12 | 3.95 ± 1.19 | 7.55 ± 2.90 |
| | MTL | 12.57 ± 9.03 | 45.79 ± 15.03 | 4.02 ± 3.79 | 5.15 ± 2.87 |
| | SANDMask | 1.49 ± 2.48 | 44.87 ± 10.27 | 2.59 ± 4.19 | 4.73 ± 7.06 |
| | SD | 6.40 ± 2.66 | 39.73 ± 14.38 | 9.43 ± 4.26 | 15.29 ± 7.22 |
| | SelfReg | 16.33 ± 6.81 | 42.06 ± 14.84 | *14.51± 8.98 | 16.61 ± 8.87 |
| | TRM | 5.00 ± 3.44 | 33.31 ± 11.40 | 7.81 ± 5.22 | 12.51 ± 7.75 |
| | VREx | 7.47 ± 4.73 | 44.50 ± 10.18 | 11.37 ± 6.68 | *17.11± 8.94 |
| | IB_ERM | 4.94 ± 2.78 | 43.01 ± 13.13 | 8.12 ± 4.46 | 13.79 ± 7.10 |
| | IB_IRM | 8.56 ± 5.49 | 44.63 ± 11.52 | 11.18 ± 6.41 | 15.06 ± 7.38 |
| BENDR | | 2.48 ± 0.91 | 28.99 ± 6.80 | 3.58 ± 1.44 | 5.79 ± 2.33 |
| Franceschi et al. | | 2.62 ± 1.02 | 44.74 ± 8.79 | 4.26 ± 2.03 | 9.65 ± 3.91 |
| SICR | | *25.34± 15.68 | 29.22 ± 20.08 | 9.19 ± 4.89 | 9.80 ± 4.41 |
| SEEG-Net | | 20.06 ± 5.56 | 32.81 ± 8.50 | 20.82± 5.70 | 22.92± 5.96 |
| PPi | | **29.76**± 5.45 | **47.59**± 5.16 | **30.92**± 3.45 | **35.51**± 2.35 |