# OpenReview forum: "PPi: Pretraining Brain Signal Model for Patient-independent Seizure Detection"
_NeurIPS.cc/2023/Conference — NeurIPS 2023 poster_

### Official Review · Reviewer_DeFx · 2023-06-26

**Soundness:** 2 fair
**Presentation:** 3 good
**Contribution:** 2 fair
**Rating:** 5
**Confidence:** 3

**Summary:**

The manuscript presents an innovative model called PPi (Pretraining-based model for Patient-independent seizure detection) for patient-independent seizure detection utilizing SEEG data. SEEG provides detailed and three-dimensional brainwave information which is advantageous for seizure detection. However, challenges emerge in modelling SEEG data due to the substantial domain shift between different patients and the evolving patterns among various brain areas.

To tackle these challenges, the authors introduce two novel self-supervised tasks during the pretraining phase. These tasks extract comprehensive information from the available SEEG data while preserving the unique characteristics of brain signals recorded from different brain areas. They also propose two techniques, channel background subtraction and brain region enhancement, to effectively address the domain shift problem between patients.

The performance of the PPi model is substantiated through extensive experiments conducted on two public datasets and a real-world clinical dataset collected by the authors. The experiments demonstrate that the PPi model outperforms state-of-the-art baselines, showcasing the practicality and effectiveness of the model. The authors also provide a visualization analysis that further validates the rationality of the two proposed domain generalization techniques.

The contributions of this work encompass the application of patient-independent seizure detection on a large-scale real-world clinical SEEG dataset under clinical requirements. This achievement highlights the innovation of the self-supervised pretraining tasks and the domain generalization techniques that the authors propose to manage the domain shift between patients. Furthermore, the superior performance of the PPi model is demonstrated through extensive experiments with other state-of-the-art methods, thereby emphasizing the significant application value of the work.


**Strengths:**

1. Figure 1. Figure 3. and Figure 4. are not only clear but also self-explanatory, demonstrating a well-designed visual depiction of the proposed method. This effectively simplifies the understanding of the framework architecture and the proposed method's workflow, serving as a useful tool for readers to follow the discussion and analysis in the manuscript.

2. The paper included extensive experimentation, which covers three different datasets. Their rigorous approach in testing the proposed method against a broad range of state-of-the-art methods provides a robust and comprehensive evaluation of its performance. Such extensive evaluation contributes to the manuscript's credibility and confirms the model's generalizability and applicability to various data contexts.

3. The performance of the PPi model, as reported in the results, is very promising. The superior performance of PPi when compared to state-of-the-art methods in seizure detection underlines the efficacy of the proposed method and its potential contribution to the field. It further exemplifies the practical significance and potential of the PPi model in real-world applications, especially in clinical scenarios.

4. The authors' inclusion of an ablation study effectively demonstrates the impact and contribution of different components of the proposed method towards the overall performance. This provides clear insights into the proposed approach's effectiveness.

**Weaknesses:**

1. In Section 4.1.2, the authors stated that their proposed self-supervised learning (SSL) task exhibits a robust capability for extracting time-domain features. However, the basis for this assertion is unclear, and the inclusion of supporting references would strengthen the argument. Furthermore, the paper does not explain why the SSL tasks are applied solely to the time domain and not the frequency domain. Providing a more detailed discussion regarding these decisions would improve the narrative and understanding of the method.

2. The clinical dataset described in Section 5.1 is highly imbalanced, yet the paper does not adequately assess how the model handles class imbalance. If the authors believe that the proposed SSL tasks can inherently address the data imbalance issue, then a more detailed analysis and validation of this claim would be beneficial.

3. The authors mention the importance of preserving unique brain area characteristics for effective seizure detection across different lesions in the introduction section. However, it remains unclear how other individual differences are accounted for in their method, beyond the variance in seizure locations. Furthermore, there is a lack of qualitative analysis supporting how well the proposed model preserves the characteristics of different brain areas across channels, which is crucial to corroborate the authors' claims.

4. Figure 5, in its current state, is relatively small and not entirely intuitive. The t-SNE plots, despite the authors mentioning that the t-SNE plots were only split after the dimensionality reduction, the plots do not clearly show that they are derived from the same calculation. I recommend reducing the total data points included in the plot, plotting both of them on the same t-SNE graph with 4 distinct colours instead of splitting them. In addition, add another set of t-SNE plots of two channels that were from different brain regions from two different patients as a comparison to demonstrate the unique characteristics of different brain regions are kept within the latent features.

5. The ablation study should also include an evaluation of the reconstruction loss term and the self-attention layers' significance. Their roles and impacts on the model's performance should be more explicitly assessed to provide a more comprehensive understanding of the model's architecture and its component's contribution.

**Questions:**

1. In Section 5.4's Ablation study, the performance of the PPi-SSL model exhibits some inconsistencies. While it outperforms PPi-SSL2 on the FNUSA dataset, it shows considerably reduced effectiveness on the Clinical dataset. Could the authors expand on the potential reasons for these discrepancies? For instance, could the model have collapsed during training in the absence of reconstruction loss? Or might the performance variation be attributed to the differing class distributions within the datasets?

2. Similarly in Section 5.4's Ablation study, it remains unclear why the condition of PPi-brain region isn't applicable to the two public datasets. Is it because the brain region enhancement cannot be used at all for the two public datasets?  A clarification on this matter from the authors would be appreciated.

**Limitations:**

please refer to the weaknesses

---

> ### Author Rebuttal · Authors · 2023-08-09
>
> We thank the reviewer for all the insightful comments. Responses to specific comments are listed below.
>
> * **W1: Add supporting references to strengthen the argument for SSL tasks.**
>
>   Thanks for the suggestions. [1], [3], [4] are some works which also apply SSL to extract time-domain features. [2] stated that the performance gap between balanced and imbalanced pre-training with SSL is significantly smaller than the gap with supervised learning, which means the SSL is more robust in our scenario.
>
>   Our SSL tasks are applied solely to the time domain because the designed tasks are mainly to capture the change of the signal over time. Especially in context swapping, this task is to enhance the coherence semantic uniqueness of contextual information, which is based on time domain information.
>
>   We will add the above discussions to our manuscript.
>
> * **W2: Add supporting references to explain why SSL can be robust to dataset imbalance.**
>
>   Thanks for the suggestions. There are some works which claim that SSL can address the data imbalance issue. For example, [1] \(SIGKDD 2023\) adopts SSL pre-training to alleviate the data imbalance. [2] (ICLR 2022 spotlight) systematically investigate SSL under dataset imbalance and find that SSL is more robust to dataset imbalance. We will add the above works to the manuscript to clarify this claim.
>
> * **W3&W4: Motivations for preserving unique brain area characteristics and additional visualization analysis as supporting.**
>
>   Thanks for the concerns and suggestions. Our response is consists of two parts.
>
>   (1) Since the similarity in structure and function of the same brain region among patients, the representations of the same brain regions can be aligned (illustrated by the case study). Thus, we only consider preserving the unique characteristics among different brain regions rather than other individuals differences.
>
>   (2) We conducted visualization analysis with 7 more examples (shown in Fig. 2 and Fig. 3 from the pdf file in global response).  As you suggested, we reduce the number of points included in the plot to a fifth of the original and plot both figures of two patients on the same t-SNE graph with 4 distinct colours.
>
>   In Fig. 2, 6 more examples were included. In each of them, the representations are from the same brain region of two different patients.
>
>   In Fig. 3, we add another example where the representations are from different brain regions of two different patients. Compared with other plots, the distributions in this plot are not totally aligned after the channel background subtraction, indicating the unique characteristics of different brain regions are preserved within the latent features.
>
> * **W5: Ablation study for reconstruction and self-attention layer.**
>
>   Thanks for the suggestion. The ablation study of the reconstruction loss term and the self-attention layers' significance is shown in Table 1 in the global response. We will add the results to the original ablation study to our revised paper, including the settings, results and corresponding analysis.
>
> * **Q1: Discussions about exhibiting performance variations on different datasets (explored by additional experiments).**
>
>   Thanks for bringing up this point. Actually on the FNUSA dataset, PPi-SSL obtains 47.29% on F2 score, which is slightly lower than PPi-SSL2 (48.42%). On MAYO and clinical dataset, the performance of PPi-SSL shows a considerably decrease compared with PPi-SSL2. In order to explore such discrepancies, we reconducted these two sets of experiments and obtained similar results:
>
>   |          | MAYO         |              |              |              | FNUSA        |              |              |              | Clinical     |              |              |              |
>   | -------- | ------------ | ------------ | ------------ | ------------ | ------------ | ------------ | ------------ | ------------ | ------------ | ------------ | ------------ | ------------ |
>   |          | Pre.         | Rec.         | F1           | F2           | Pre.         | Rec.         | F1           | F2           | Pre.         | Rec.         | F1           | F2           |
>   | PPi-SSL2 | 41.27+/-5.59 | 39.10+/-5.43 | 36.02+/-4.97 | 36.96+/-5.13 | 50.45+/-7.45 | 49.43+/-6.86 | 47.25+/-2.80 | 48.78+/-4.49 | 18.94+/-4.82 | 47.21+/-6.87 | 16.85+/-3.02 | 24.15+/-3.16 |
>   | PPi-SSL  | 33.23+/-8.44 | 24.02+/-5.12 | 26.12+/-5.83 | 24.47+/-5.24 | 74.20+/-7.96 | 40.11+/-5.78 | 52.56+/-4.93 | 45.36+/-5.35 | 9.87+/-2.14  | 25.32+/-4.65 | 12.74+/-2.66 | 16.19+/-2.99 |
>
>   So we think the performance variation might be attributed to the differing class distributions within the datasets.
>
> * **Q2: Clarification about brain region enhancement task for the two public datasets.**
>
>   In the manuscript, line 296: "For the ablation experiments on two public datasets, brain region enhancement is not applied due to lack of information about brain regions."
>
>   As the brain region enhancement needs the brain region labels and these two public datasets do not contain brain region labels, brain region enhancement cannot be applied to the two public datasets. We will reorganize our description in the text to make it more clear: "Due to the requirement of brain region labeling in brain region enhancement, the lack of such labels in the two public datasets makes brain region enhancement inapplicable to the public datasets".
>
> References:
>
> [1]Chen J, Yang Y, Yu T, et al. Brainnet: Epileptic wave detection from seeg with hierarchical graph diffusion learning[C]. 2022.
>
> [2]Liu H, HaoChen J Z, Gaidon A, et al. Self-supervised learning is more robust to dataset imbalance[J]. 2021.
>
> [3]Shao Z, Zhang Z, Wang F, et al. Pre-training enhanced spatial-temporal graph neural network for multivariate time series forecasting[C]. 2022.
>
> [4]Nie Y, Nguyen N H, Sinthong P, et al. A time series is worth 64 words: Long-term forecasting with transformers[J]. 2022.

---

> > ### Comment · Reviewer_DeFx · 2023-08-16
> >
> > Thanks for addressing my comments.
> > I have increased the score

---

> > > ### Author Response · Authors · 2023-08-17
> > > **Thanks for the reviews**
> > >
> > > We truly appreciate your effort in improving our paper and recognition of our work.

---

### Official Review · Reviewer_LsDA · 2023-07-02

**Soundness:** 3 good
**Presentation:** 3 good
**Contribution:** 2 fair
**Rating:** 6
**Confidence:** 4

**Summary:**

This paper proposes a model called PPi that pre-trains on SEEG data using two self-supervised tasks, followed by a channel background subtraction step as well as a brain region enhancement task for patient-independent seizure detection. Experiments on two public datasets and an internal dataset suggest that PPi outperforms existing models for seizure detection. Visualization of latent representations using a t-SNE plot indicate the effectiveness of the channel background subtraction step.

**Strengths:**

1. There is some originality in the design of the pre-training tasks and the channel background subtraction step.
2. Overall the methods are sound and the paper is relatively easy to understand.
3. The proposed model PPi shows improved performance over a set of baselines.

**Weaknesses:**

1. Several design choices in Methods need more justifications (see my questions below).
2. All the datasets are from a small number of patients, which could be a limitation and should be discussed.
3. Some notations and equations in Methods are not necessary, and can be better explained using plain texts. Overloaded notations make it harder to understand. For instance, Definition 1 is not necessary.
4. While PPi outperforms baselines, the overall performance of PPi is still low, particularly on the clinical dataset, which could limit its clinical applicability.

**Questions:**

1. Some notations and equations in Methods are not necessary (e.g., Definition 1). Replacing some notations with plain language can help readability.
2. In the pre-training phase, how to sample contexts if the target segment is in the beginning or end of the signal? Please clarify.
3. More justifications for design choices are needed in Methods. For example, in “Channel discrimination” section, why is the difference vector used instead of a concatenated vector as done in “context swapping”? Why and how is self-attention used to aggregate frequency and time-domain representations?
4. Reconstruction loss is included in the channel discrimination pre-training task. Please include an ablation to show the impact of reconstruction loss.
5. In Section 5.4, the authors state that “for the ablation experiments on two public datasets, brain region enhancement is not applied due to lack of information about brain regions.” Does this mean that the results in Table 1 do not include brain region enhancement task for the two public datasets? Please clarify.
6. For the visualization analysis in Section 5.5, it would be good to show other patient examples to make sure that the observation is consistent across patients (this should be doable given that there is only 7 patients in the clinical dataset).
7. In my opinion, it’s a limitation that the data come from a small number of patients. This limitation should be discussed.

**Limitations:**

Some limitations are discussed in the end of the paper.

---

> ### Author Rebuttal · Authors · 2023-08-09
>
> We thank the reviewer for all the insightful comments. Responses to specific comments are listed below.
>
> * **Q1: Replace some notations with plain language in the manuscript.**
>
>   Thank you for this good suggestion. The plain language version for Definition 1 is in line118-119: "Our goal is to utilize the data of labeled patients (source domains) to train a model which can be directly adopted to the data of unseen patients (target domains)." We will remove Definition1 for better readability.
>
>   In line 160: "where the sampling probability $P(c1= c2) = P(c1\neq c2) = 0.5$", we will change with "where the two sequences are sampled from the same or different channels with equal probability."
>
> * **Q2: The solution of sampling contexts for the beginning or end of the signal.**
>
>   In the pre-training phase, the data scale is large, so we just simply ignore the beginning and the end of the signal. We apologize for not clearly clarifying this and we will add the clarification to the main manuscript.
>
> * **Q3: Clarification for "difference vector" used in "channel discrimination" and ablation for aggregation strategy.**
>
>   In channel discrimination, since the target is to compare the difference between two signals, we consider to use a elementwise absolute difference. This choice is motivated from [1], in which they do a relative positioning (RP) task. In RP, they train the model to dicriminate whether the given two time windows are far or close to each other. They use the elementwise absolute difference to combine the two vectors. Since RP and channel discrimination both involve the comparison of the difference between the given two samples, we think a elementwise absolute difference may be more suitable.
>
>   While context swapping does not involve a direct comparison of the difference between the given two samples, but more focus on the coherence semantic uniqueness of contextual information. As a result, we think a concatenated vector is more suitable for context swapping.
>
>   In the aggregation strategy, by assigning weights to representations in the time and frequency domains through the attention mechanism, the model can adaptively choose between the two representations. In the implementation, we feed the time domain representation and frequency domain representation into a self-attention layer, and then apply a mean pooling to the output of this layer. To verify the effectiveness of self-attention experimentally, we add an addtional ablation experiment on self-attention (result is shown in Table 1 in the global response).
>
> * **Q4: Ablation for reconstruction loss.**
>
>   Thanks for the good advice. The ablation with the reconstrcution loss is shown in Table 1 in the global response. We will add this result along with the ablation for self-attention aggregation to the original ablation study to our revised paper, including the settings, results and corresponding analysis.
>
> * **Q5: Clarification about brain region enhancement task for the two public datasets.**
>
>   Yes, the results of the two public datasets in Table 1 do not include brain region enhancement task. As the brain region enhancement needs the brain region labels and these two public datasets do not contain brain region labels, brain region enhancement cannot be applied to the two public datasets.
>
> * **Q6: Additional experiments for case study.**
>
>   Thank you for this good suggestion. We conduct another 6 groups of visualization analysis and show the results in Fig. 2 from the pdf file in global response. The patients we chose cover all the 7 patients in the clinical dataset. In the new visualization analysis, we made the following improvements to make the results more clear (suggestions for improvement come from reviewer DeFx):
>
>   (1) We reduce the number of points included in the plot to a fifth of the original by random sampling.
>
>   (2) We plot the representations of two patients on the same t-SNE graph with 4 distinct colours instead of splitting them.
>
>   Overall, in each group of experiments, channel background subtraction shows a similar effect, indicating that the effect of channel background subtraction is consistent across patients.
>
> * **Q7: Discussions about subject numbers.**
>
>   Thanks for pointing this out. The number of patients in our datasets (7 in clinical dataset, 13 in MAYO and 18 in FNUSA) are relatively small compared to other fields (e.g. EEG). This is a good suggestion and we will discuss this limitation in our manuscript.
>
>   Actually SEEG is an emerging field, and the conditions for obtaining data are very strict as it requires craniotomy surgery for electrode implantation and long-term data recording. Some other works in this field also contains a limited number of patients (e.g., [2] contains 10 patients, [3] contains 10 patients). We are working hard to communicate with hospitals, hoping to release more data. Strive to alleviate the problem of insufficient data in the field of SEEG.
>
>
> References:
>
> [1]Hubert Banville, Omar Chehab, Aapo Hyv.rinen, Denis-Alexander Engemann, and Alexandre Gramfort. Uncovering the structure of clinical eeg signals with self-supervised learning. Journal of Neural Engineering, 18:046020, 2021.
>
> [2]Chen J, Yang Y, Yu T, et al. Brainnet: Epileptic wave detection from seeg with hierarchical graph diffusion learning[C]//Proceedings of the 28th ACM SIGKDD Conference on Knowledge Discovery and Data Mining. 2022: 2741-2751.
>
> [3]Wang C, *et al.*, "BrainBERT: Self-supervised representation learning for intracranial recordings." *International Conference on Learning Representations, 2023*.

---

> > ### Comment · Reviewer_LsDA · 2023-08-13
> >
> > Thank you for addressing my comments. I have now increased the score.

---

> > > ### Author Response · Authors · 2023-08-15
> > > **Thanks for the reviews**
> > >
> > > We are truly grateful for the reviewer’s feedback and recognition of our efforts.

---

> > > > ### Comment · Reviewer_LsDA · 2023-08-19
> > > >
> > > > Dear Authors,
> > > >
> > > > I want to raise another concern that while this paper focuses on SSL, surprisingly, the authors do not discuss or compare their methods to any of the prior works on SSL for EEG/biosignals/time series. I’d like to see a section discussing prior SSL approaches in the “Related Work” section, as well as how the proposed SSL approaches compare to prior SSL approaches in terms of model performance.
> > > >
> > > > Example prior works on SSL for biosignals include, but not limited to:
> > > >
> > > > 1.	Banville, H., Chehab, O., Hyvärinen, A., Engemann, D.-A. & Gramfort, A. Uncovering the structure of clinical EEG signals with self-supervised learning. J. Neural Eng. 18, (2021).
> > > > 2.	Mohsenvand, M. N., Izadi, M. R. & Maes, P. Contrastive Representation Learning for Electroencephalogram Classification. in Proceedings of the Machine Learning for Health NeurIPS Workshop (eds. Alsentzer, E. et al.) vol. 136 238–253 (PMLR, 2020).
> > > > 3.	Kostas, D., Aroca-Ouellette, S. & Rudzicz, F. BENDR: Using Transformers and a Contrastive Self-Supervised Learning Task to Learn From Massive Amounts of EEG Data. Front. Hum. Neurosci. 15, 653659 (2021).
> > > > 4.	Franceschi, J.-Y., Dieuleveut, A. & Jaggi, M. Unsupervised Scalable Representation Learning for Multivariate Time Series. in Advances in Neural Information Processing Systems (eds. Wallach, H. et al.) vol. 32 (Curran Associates, Inc., 2019).
> > > > 5.	Tang, S. et al. Self-Supervised Graph Neural Networks for Improved Electroencephalographic Seizure Analysis. in International Conference on Learning Representations (2022).

---

> > > > > ### Author Response · Authors · 2023-08-21
> > > > > **Response to the new comment**
> > > > >
> > > > > Thanks for the good suggestions. The target of our work is conducting seizure detection for **SEEG data** on a **domain generalization (DG)** setting. When considering the comparison methods, we first choose the DG method on SEEG which is the most relevant to our work. For further comparison, we also select other DG methods which are designed for EEG or more general fields. We apologize for not considering the SSL-based methods as baselines. We hope the following response can address your concern.
> > > > >
> > > > > * Firstly, we will add the section below which discusses prior SSL approaches in the “Related Work”:
> > > > >
> > > > >   "Self-supervised learning is an effective approach when the labeled data is limited. In the field of neural signal (e.g. SEEG, EEG), the label is often hard to obtain. Thus, researchers have developed some SSL methods for this field. Banville et al.[1] utilize relative positioning, temporal shuffling and contrastive predictive coding as the pretext tasks for EEG. Mohsenvand et al.[2] and Kostas et al.[3] model EEG signal using contrastive learning. Franceschi et al.[4] propose an unsupervised method to learn universal embeddings of time series. Tang et al.[5] construct a graph to model EEG data with self-supervised learning. However, these works do not explicitly align the distribution gaps between different domains, which is crucial under the DG setting, especially for data with large domain differences such as SEEG."
> > > > >
> > > > > * Secondly, we conduct experiments for your given works on our clinical dataset which better reflects the model performance in clinical applications. For the given five works, [5] employs a graph-based approach that generates graph nodes by utilizing channels, with each channel representing a node. In SEEG, the number of channels differs across patients, limiting the applicability of this method to individual patients only. Thus it does not satisfy our patient-independent setting.
> > > > >
> > > > >   Due to the approaching deadline of the discussion period and the authors of [1] and [2] do not make the source code publicly avaiable, we conduct experiments on [3] and [4] on our clinical dataset. For [1] and [2], we will try our best to reproduce these works and add the results in the final version of our paper, including the settings, results and corresponding analysis. The results of [3] and [4] are shown below:
> > > > >
> > > > >   |                   | Clinical     |              |              |              |
> > > > >   | ----------------- | ------------ | ------------ | ------------ | ------------ |
> > > > >   |                   | Pre.         | Rec.         | F1           | F2           |
> > > > >   | BENDR             | 2.48+/-0.91  | 28.99+/-6.80 | 3.58+/-1.44  | 5.79+/-2.33  |
> > > > >   | Franceschi et al. | 2.62+/-1.02  | 44.74+/-8.79 | 4.26+/-2.03  | 9.65+/-3.91  |
> > > > >   | PPi               | 29.76+/-5.45 | 47.59+/-5.16 | 30.92+/-3.45 | 35.51+/-2.35 |
> > > > >
> > > > >   Since these works do not explicitly align the distribution gaps between different domains, under patient-independent setting, their performances on our clinical dataset are relatively low compared to our method and some well-performed DG-based methods.
> > > > >
> > > > > Thanks again for your valuable suggestions which greatly improve our work.
> > > > >
> > > > > References:
> > > > >
> > > > > [1]Banville, H., Chehab, O., Hyvärinen, A., Engemann, D.-A. & Gramfort, A. Uncovering the structure of clinical EEG signals with self-supervised learning. J. Neural Eng. 18, (2021).
> > > > >
> > > > > [2]Mohsenvand, M. N., Izadi, M. R. & Maes, P. Contrastive Representation Learning for Electroencephalogram Classification. in Proceedings of the Machine Learning for Health NeurIPS Workshop (eds. Alsentzer, E. et al.) vol. 136 238–253 (PMLR, 2020).
> > > > >
> > > > > [3]Kostas, D., Aroca-Ouellette, S. & Rudzicz, F. BENDR: Using Transformers and a Contrastive Self-Supervised Learning Task to Learn From Massive Amounts of EEG Data. Front. Hum. Neurosci. 15, 653659 (2021).
> > > > >
> > > > > [4]Franceschi, J.-Y., Dieuleveut, A. & Jaggi, M. Unsupervised Scalable Representation Learning for Multivariate Time Series. in Advances in Neural Information Processing Systems (eds. Wallach, H. et al.) vol. 32 (Curran Associates, Inc., 2019).
> > > > >
> > > > > [5]Tang, S. et al. Self-Supervised Graph Neural Networks for Improved Electroencephalographic Seizure Analysis. in International Conference on Learning Representations (2022).

---

### Official Review · Reviewer_M3UK · 2023-07-05

**Soundness:** 3 good
**Presentation:** 3 good
**Contribution:** 3 good
**Rating:** 8
**Confidence:** 3

**Summary:**

Seizure detection using stereoencephalographic data is crucial for epilepsy diagnosis. Owing to the large variety of seizure patterns and pathology, automated seizure detection is quite challenging, and manual annotation of data remains necessary. This study proposes a model to detect seizures from SEEG data in a patient-independent manner (PPi). The proposed model is based on self-supervised learning tasks. To handle the frequency domain shift between patients, they proposed background channel subtraction and brain region enhancement techniques.
The model presented in this article seems to outperform existing models, especially on clinical data. This article presents a model that could potentially help clinicians detect seizures.


**Strengths:**

This study attempts to solve a common issue in epilepsy diagnosis, but this could also be helpful in annotating large SEEG datasets. The presentation of the results is clear, and all the results appear to be presented. The main strength of the proposed model is its capacity to be applicable to all types of patients by dealing with the huge domain shift between patients. The two techniques of channel background subtraction and brain region enhancement seem to be highly innovative and could potentially be applied to other datasets to answer other questions. The model performance appears to be very reasonable, especially for clinical data.
One of the main strengths of this study is the use of a self-supervised learning method that allows non-annotated data to be obtained, particularly when working on such rare datasets.

**Weaknesses:**

One of the main issues in this study is that they do not show raw traces of data. It could have been good that we see some of the automatically detected seizures to see whether the well-detected seizures are the easiest to detect or if some very complex seizures are also detected. The main problem with epileptic seizures is that their patterns differ depending on the pathology and brain region. Some seizures, such as low-voltage fast-activity seizures, could potentially be very hard to detect by a model, at least the real start of the seizures.
They could be precise in the article if, for the seizures detected, there were many manual adjustments to do afterwards to correctly place the start of the seizure.
The performance of the model is correct based on other existing models; however, the presented results are not sufficient to fully rely on automatic detection. As they said in the discussion, this could be helpful for clinicians, but it cannot replace manual annotation at the moment.


**Questions:**

What preprocessing steps did you apply to the dataset ? However, this was not clear in the article.


Did you process any artifact rejection before training ?


Can you please show the raw traces of EEG for well-detected and undetected seizures ? Are there some patterns that the model is unable to detect ?


**Limitations:**

The authors seems to correctly adress the limitations of the article

---

> ### Author Rebuttal · Authors · 2023-08-09
>
> We thank the reviewer for all the insightful comments. Responses to specific comments are listed below.
>
> * **Q1&Q2: Preprocessing steps for the datasets.**
>
>   For the public datasets, we first remove the power line noise and then dowm sample the data to 500Hz. For the clinical dataset, we remove the powerline noise, down sample the data to 250Hz and apply a normalization to each channel by the equation $\frac{x-\mu}{\sigma}$, where $\mu, \sigma$ are the mean and standard deviation of the channel.
>
>   For artifact rejection, we remove the power line noise in the public datasets and clinical dataset before training.
>
>   We apologize for not clearly mentioning the preprocessing steps and will add the above description to our manuscript.
>
> * **Q3: Examples of SEEG signals of well-detected and undetected seizures by the model.**
>
>   Of course, we pick up the raw traces of well-detected and undetected seizures which are shown in Fig. 1 in the pdf file in global response. In Fig. 1, we show two examples. In the examples, we highlight the normal signals (when the patient is in the absence of epilepsy) with yellow, the well detected seizures with red, and the undetected seizures with blue.
>
>   In most cases, those more pronounced seizure signals (with more violent fluctuations, highlighted in red) were easier for the model to identify. However, some seizure signals are very similar to their nearby normal signals (highlighted in blue), making it difficult for the model to identify such signals.

---

> > ### Comment · Reviewer_M3UK · 2023-08-17
> >
> > Thank you for your answer and the clarifications.

---

> > > ### Author Response · Authors · 2023-08-18
> > > **Thanks for the reviews**
> > >
> > > We genuinely appreciate your valuable reviews and recognition of our work.

---

### Official Review · Reviewer_iQPF · 2023-07-06

**Soundness:** 3 good
**Presentation:** 2 fair
**Contribution:** 3 good
**Rating:** 7
**Confidence:** 3

**Summary:**

This article deals with domain shifts in seizure detection with stereoelectroencephalography (SEEG), an emerging acquisition method in this field. To tackle this problem, the authors propose two different self-supervised tasks to learn meaningful features from the SEEG and propose also two preprocessing techniques to reduce the shift between the domain. All this propositions allow to beat the SOTA methods on two public datasets and also on their own clinical dataset.

**Strengths:**

This paper proposes a new PPi method to be applied to an emerging field SEEG. PPi shows very good results compare to what the SOTA proposes. Moreover, the research is done in coordination with doctors, so the paper deals with real-world clinical issues.

The method is very clear.

The paper implements very nice experiments, allowing us to see the benefits of the method and the effects on the data.

The comparison is made over several other methods.

In addition, a new clinical dataset was collected for this paper. SEEG is a new field where every dataset is crucial to improve the impact in the science community.

**Weaknesses:**

In the experimental part, the authors' claims do not link to the performance in Table 1. For example, they claim 54.93% on F2 score of improvement on the clinical dataset, but the average performance is only 35.51% in Table 1.

The PPi method is a concatenation of several components (SSL, frequency spectral features, brain region ...). Even with the ablation study is hard to understand what is the improvement of each component. The last experiment proposes visualizing the effect of the channels' background subtraction, but it is the only visible effect. Each other component does not seem to improve the results much except when all the components are combined.

**Questions:**

The paper claims a 54% improvement in the F2 score, but it seems that Table 1 gives other results. 35,51% of F2 score is only a 17.40% improvement if we compare it to the best baseline results (i.e. MiniRocket with VREx). From where does this claim come from?

In Table. 2 for the ablation study, It is hard to understand which layer is removed. For example, PPi-power means the all PPi framework without feature and spectral power, so only encoding features? same question for PPi-SSL. With this experiment, it seems that none of the layers are useful alone to improve performance. For each row of the tables, the performances are below the baseline. is PPi useful only when all the components are taken?

Proposing a new clinical dataset can be game-changing in such an emerging field as SEEG. Will the dataset be publicly available too?

The score on the clinical dataset seems worse than on the public dataset. Do you think the score can be improved? Does 35% of F2 score enough to help the doctor to detect seizures in the brain?






**Limitations:**

No limitation

---

> ### Author Rebuttal · Authors · 2023-08-09
>
> Thank you for your constructive and detailed comments. Responses to specific comments are listed below.
>
> * **Q1: Calculation method of performance improvement value in the manuscript.**
>
>   We apologize that the calculation of the improvement may not be clearly stated in the manuscript. Actually, the improvement claimed in our manuscript is a relative improvement which is also utilized in [1]. Specifically, suppose the performances are $a$% and $b$% (a>b), then the relative improvement is calculated as $\frac{a-b}{b}\times 100$%. We will add a clear description in the main manuscript.
>
> * **Q2: Detailed explanation and discussions about ablation study.**
>
>   In the ablation study, PPi-SSL1, PPi-SSL2, PPI-SSL are included to demonstrate the effectiveness of our self-supervised tasks. PPi-SSL1 means in the pre-training stage, we only do the context swapping task. Likewise, PPi-SSL2 means in the pre-training stage, we only do the channel discrimination task. **PPi-SSL means the encoder is not pre-trained by the self-supervised tasks and directly trained in the downstream seizure detection.** PPi-power is included to demonstrate the effectiveness of the features from frequency domain. **PPi-power means we only use the encoder pre-trained with the self-supervised tasks to encode the data to representations, but do not use the power spectral density.**
>
>   Since each component is carefully designed, if any part is missing, the performance of the model will drop a lot compared to the complete version, resulting in a lower than the best baseline. It also reflects that our model is not simply a patchwork of individual components, but that these components work together.
>
> * **Q3: Plans about releasing our clinical dataset.**
>
>   Thank you for providing this good advice. Releasing such a clinical dataset involves highly sensitive intracranial data and personal privacy concerns and we are working hard to make the dataset publicly available.
>
>   It is very difficult to make this dataset all public in a short time. Therefore, we plan to promote the release of our dataset in the following two steps:
>
>   * Step1: We will engage in proactive communication with the hospital and strive to make a subset of raw data available within a half year. Similar to the procedures outlined in our manuscript, these data releases will also adhere to ethical review mandates.
>   * Step2: In the future, we will examine the feasibility of releasing the complete dataset once it receives the necessary ethical review approval. This will enable researchers to utilize the large-scale dataset for further research purposes.
>
> * **Q4: Discussions about the potential improvement.**
>
>   In addition to the model, the dataset itself will also have a great impact on the score (such as the ratio of positive and negative samples, data noise, label noise, etc.). So in the experiments, the scores on clinical dataset are lower than those on public datasets. We believe that scores can be improved mainly in two ways. The first is optimizing the model (such as model design, model training) to get a higher performance. The second is to improve the quality of the dataset (e.g., clean the data carefully before training). We will continue to explore for better performance in the future.
>
>   In the main manuscript (limitations and future works), we stated that the predicted results of our model are mainly serve as a reference to assist doctors to achieve more efficient clinical diagnosis and treatment, rather than completely replace doctors in seizure detection. Additionally, during our research, we engaged in discussions with doctors who expressed their positive response towards our model’s performance that aids in the diagnosis of epilepsy at the current stage.
>
> References:
>
> [1]Chen J, Yang Y, Yu T, et al. Brainnet: Epileptic wave detection from seeg with hierarchical graph diffusion learning[C]//Proceedings of the 28th ACM SIGKDD Conference on Knowledge Discovery and Data Mining. 2022: 2741-2751.

---

> > ### Comment · Reviewer_iQPF · 2023-08-18
> >
> > I thank the authors for the answer of my questions and the clarifaction.

---

> > > ### Author Response · Authors · 2023-08-19
> > > **Thanks for the reviews**
> > >
> > > We are grateful for your valuable reviews and recognition of our work.

---

### Official Review · Reviewer_zvti · 2023-07-09

**Soundness:** 2 fair
**Presentation:** 2 fair
**Contribution:** 2 fair
**Rating:** 5
**Confidence:** 4

**Summary:**

This paper proposes a patient-independent seizure detection framework called PPi for stereoelectroencephalography (SEEG) data. It utilizes self-supervised learning for taking into account discriminability of brain areas and contextual coherence of SEEG signals to preserve the patterns of different channels. The authors propose to use channel background subtraction for distributional alignment of brain regions across patients and handle inter-patient domain shift. Evaluation is performed on two public and one clinical data-set against several baseline methods.

**Strengths:**

1. Evaluation is performed on three separate datasets against several baseline methods.

2. Performance tables reported appear to consistently suggest that PPi provides improvements against the selected baselines for seizure detection.

3. The ablation studies support the need for different components that the model combines.


**Weaknesses:**

1. The presentation of the main methodology reads as a piecing together of multiple existing modules making the novelty of the proposed method unclear for this application.

2. The tables do not report standard deviation measures for any of the methods or discussions of statistical significance measures, making it hard to gauge the robustness of the method to the data splits.



**Questions:**

1. The channel distribution alignment visualization in Fig. 5 does not appear to result in a clear separation between the epileptic and controls subjects. Could the authors provide more intuition on why the representation leads to enhanced classification performance between the two classes?

2. A suggestion would be to include a shortened version of the explanation Section E: Brain Region Enhancement Details from the appendix in the main text of the work to improve readability and aid readers in understanding the multi-classification setup.

**Limitations:**

The authors have identified the limitations of their work in a separate subsection within the conclusion and have indicated potential for clinical deployment. It would be great if the authors could provide more information regarding the clinical targets (eg. detection accuracy) to be met for translation.

---

> ### Author Rebuttal · Authors · 2023-08-09
>
> We thank the reviewer for all the insightful comments. Responses to specific comments are listed below.
>
> * **W1: Clarification for the unclearness of novelty raised by presentation.**
>
>   Thanks for pointing this out. The novelty of the proposed method are listed below:
>
>   * Our method contains two novel self-supervised pretraining tasks (i.e., channel discrimination and context swapping). Different from other SSL studies, the motivation of our designed tasks is to preserve the unique patterns of each channel, which is more consistent with the physiological mechanism of seizures.
>
>   * We propose two techniques including channel background subtraction and brain region enhancement in our method to handle the domain shift between different patients. Different from other domain generalization works (e.g. [1], [2], [3] ), we fully consider the characteristics of SEEG in our design, so that we can achieve better performance in this scenario.
>
>   We will incorporate the above content into the methodology section to enhance the clarity of the novelty in the proposed method.
>
> * **W2: Report standard deviation of the experiments.**
>
>   We are sorry for not reporting the standard deviation of the experiments. Due to the limitation of the reply characters, we report the results with standard deviation of our model and several well-performed baselines as follows:
>
>   |                 | MAYO         |         |        |           | FNUSA        |               |              |              | Clinical     |               |              |              |
>   | --------------- | ------------ | ------------- | ------------ | ------------ | ------------ | ------------- | ------------ | ------------ | ------------ | ------------- | ------------ | ------------ |
>   |                 | Pre.         | Rec.          | F1           | F2           | Pre.         | Rec.          | F1           | F2           | Pre.         | Rec.          | F1           | F2           |
>   | TCN+CDANN | 16.13+/-5.80 | 68.14+/-4.17  | 24.74+/-7.67 | 37.84+/-8.74 | 32.71+/-6.42 | 69.62+/-2.77  | 43.09+/-5.61 | 54.88+/-3.58 | 1.42+/-0.51  | 44.20+/-13.85 | 2.72+/-0.98  | 6.08+/-2.21  |
>   | TCN+MLDG        | 18.88+/-4.22 | 61.38+/-4.65  | 27.96+/-4.92 | 40.39+/-4.78 | 32.81+/-6.61 | 56.81+/-2.41  | 40.41+/-5.43 | 48.24+/-3.71 | 28.85+/-9.34 | 6.68+/-3.92   | 2.55+/-0.58  | 2.58+/-0.89  |
>   | MiniRocket+MTL  | 21.67+/-7.15 | 46.11+/-12.08 | 27.72+/-7.38 | 35.03+/-7.90 | 56.85+/-5.25 | 59.71+/-10.71 | 56.93+/-6.94 | 58.28+/-9.00 | 12.57+/-9.03 | 45.79+/-15.03 | 4.02+/-3.79  | 5.15+/-2.87  |
>   | MiniRocket+VREx | 38.63+/-7.60 | 33.92+/-8.63  | 33.37+/-6.66 | 32.88+/-7.15 | 65.23+/-9.83 | 55.20+/-9.33  | 54.34+/-5.36 | 53.91+/-7.58 | 7.47+/-4.73  | 44.50+/-10.18 | 11.37+/-6.68 | 17.11+/-8.94 |
>   | SEEG-Net       | 45.41+/-9.96 | 45.62+/-9.56  | 43.54+/-8.84 | 44.22+/-8.98 | 69.39+/-9.23 | 53.75+/-7.62  | 60.02+/-8.05 | 55.99+/-7.73 | 20.06+/-5.56 | 32.81+/-8.50  | 20.82+/-5.70 | 22.92+/-5.96 |
>   | PPi | 49.85+/-6.93 | 69.67+/-2.82  | 54.35+/-4.72 | 61.07+/-4.69 | 71.73+/-4.06 | 70.81+/-2.14  | 70.61+/-2.82 | 70.55+/-2.28 | 29.76+/-5.45 | 47.59+/-5.16  | 30.92+/-3.45 | 35.51+/-2.35 |
>
>   We will **update all the experiment results** in the manuscript with standard deviation in the final version.
>
> * **Q1: Clarification for Fig.5 in case study with examples.**
>
>   Thanks for pointing this out. In the field of SEEG for seizure detection, there will be some ambiguous samples in the labeling process of epilepsy. Not even human experts can be 100% sure about the labels of these samples, thus there will be some discrepancies in the results marked by different annotators in the labeling process.  It is also difficult for the model to handle these samples. For exmaple, in Fig. 1 from the pdf file in global response, the signals highlighted with blue are very similar with the normal signals (highlighted with yellow) but were labeled as seizure. Likewise, there will also be some signals that look like seizures but are labeled as normal.
>
>   Due to the above reasons, in our scenario, it is almost impossible to result in a very clear seperation between seizure and normal samples. In Fig. 5 of the manuscript, the seizure samples are clustered at the edges, enabling the classifier to distinguish most of the samples.
>
> * **Q2: Add a shortend version of Section E in the appendix to the main text.**
>
>   Thank you for the good suggestion. We will add a new section (Section 5.6) in the manuscript which includes one of the figures from Fig. 7 to Fig. 13 in Appendix and the following paragraph:
>
>   "In order to demonstrate the effectiveness of brain region enhancement, we calculate the confusion matrix of the multi-classification. Fig. 6 shows the confution matrix from one of the patients (the confusion matrix of all the patients are shown in Appendix E), in which the vertical axis represents the multi-class label $y_{c,k}'$ and the horizontal axis represents the multi-class prediction result $\hat{y}_{c,k}'$. We can see that in the confusion matrix, most samples are distributed on the main diagonal, which reflects the good performance of the multi-classification task, illustrating the effectiveness of brain region enhancement."
>
> References:
>
> [1]Yiping Wang, Yanfeng Yang, Gongpeng Cao, Jinjie Guo, Penghu Wei, Tao Feng, Yang Dai, Jinguo Huang, Guixia Kang, and Guoguang Zhao. Seeg-net: An explainable and deep learning based cross-subject pathological activity detection method for drug-resistant epilepsy. Computers in Biology and Medicine, page 105703, 2022.
>
> [2]Daehee Kim, Youngjun Yoo, Seunghyun Park, Jinkyu Kim, and Jaekoo Lee. Selfreg: Self-supervised contrastive regularization for domain generalization. In ICCV, pages 9619–9628, 2021.
>
> [3]David Krueger, Ethan Caballero, Joern-Henrik Jacobsen, Amy Zhang, Jonathan Binas, Dinghuai Zhang, Remi Le Priol, and Aaron Courville. Out-of-distribution generalization via risk extrapolation (rex). In ICML, pages 5815–5826. PMLR, 2021.

---

> > ### Comment · Reviewer_zvti · 2023-08-14
> > **Response to the Rebuttal**
> >
> > Thank you for responding to my review comments. Based on the rebuttal, I have increased my score by a point.

---

> > > ### Author Response · Authors · 2023-08-15
> > > **Thanks for the reviews**
> > >
> > > We sincerely appreciate the reviewer’s effort in helping us to enhance the paper and recognition of our work.

---

### Official Review · Reviewer_oXnd · 2023-07-16

**Soundness:** 3 good
**Presentation:** 3 good
**Contribution:** 3 good
**Rating:** 6
**Confidence:** 5

**Summary:**

This paper presents a pretraining-based model for patient-independent seizure detection (PPi) on SEEG data in the clinical scenario. The proposed method adopts a self-supervised pretraining strategy to extract information from SEEG signals while preserving the unique characteristics of each channel, and applies channel background subtraction and brain region enhancement techniques to improve the generalization ability of PPi. The experimental results carried out on two public datasets and a real clinical dataset have shown that the proposed method outperforms the SOTA baselines.

**Strengths:**

- A pretraining-based model for patient-independent seizure detection (PPi) using SEEG signals.
- The proposed method adopts two self-supervised pretraining strategies (channel discrimination and context swapping) to extract information from SEEG signals while preserving the unique characteristics of each channel, and applies channel background subtraction and brain region enhancement techniques to effectively tackle the domain shift problem and thus improve the generalization ability of PPi.

**Weaknesses:**

-  The use of PSD-based features which is widely applied in the literature to extract features from the frequency domain of the EEG/SEEG signals.


**Questions:**

1) Various types of EEG/SEEG features have been proposed in the literature. It is suggested to highlight the effectiveness of PSD-based features in the proposed method.
2) Update the references by considering some recent and relevant studies.


**Limitations:**

Yes, but briefly. It is suggested to discuss more this point.

---

> ### Author Rebuttal · Authors · 2023-08-09
>
> We thank the reviewer for all the insightful comments. Responses to specific comments are listed below.
>
> * **Q1: Highlight the effectiveness of PSD-based features in the proposed method.**
>
>   Thank you for the good suggestion. There are some related works which support the effectiveness of PSD-based method in seizure deteciton. For example, [1] stated that the spectral power of brain signal has the ability to track the transient changes before and during seizure.  [2] argued that the spectral power in certain sub-bands of the SEEG, specifically in higher frequency sub-bands, may play a key role in seizure prediction.
>
>   We will add the above works in the manuscript to highlight the effectiveness of PSD-based features in our proposed method.
>
> * **Q2: Update the references.**
>
>   Thank you for the good suggestion. We will update some of the references to more recent studies. The updates are as follows:
>
>   - In line 31 of the manuscript, sentence "After collecting the SEEG recordings of patients, the process of epilepsy detection and diagnosis is traditionally treated as a manual task that highly depends on a few experienced neuroscientists, requiring considerable time and human resources". We will replace the original reference of this sentence by [5].
>
>   - In line 38 of the manuscript, sentence "However, existing works for SEEG-based seizure detection mainly focus on the patient-specific setting". We will replace the original reference of this sentence by [3].
>
>   * In line 82 of the manuscript, sentence "Ayoubian et al. employ wavelet decomposition, feature extraction, adaptive thresholding and artifact removal in SEEG data". We will replace the sentence by "Truong et al. [6] propose Integer-Net to conduct seizure detection on both EEG and SEEG".
>
>   * In line 101 of the manuscript, we will add a sentence "Zhang et al. [4] regularize the discrepancy between closely-related domains to achieve domain generalization".
>
> References:
>
> [1]M. Bandarabadi, C. A. Teixeira, J. Rasekhi, and A. Dourado, “Epileptic seizure prediction using relative spectral power features,” Clin Neurophysiol., 2014.
>
> [2]Netoff T, Yun P, Parhi K. Seizure prediction using cost-sensitive support vector machine. Engineering in medicine and biology society, 2009 EMBC 2009. In: Annual international conference of the IEEE 2009. p. 3322–5.
>
> [3]Chen J, Yang Y, Yu T, et al. Brainnet: Epileptic wave detection from seeg with hierarchical graph diffusion learning[C]//Proceedings of the 28th ACM SIGKDD Conference on Knowledge Discovery and Data Mining. 2022: 2741-2751.
>
> [4]Zhang W, Ragab M, Foo C S. Domain Generalization via Selective Consistency Regularization for Time Series Classification[C]//2022 26th International Conference on Pattern Recognition (ICPR). IEEE, 2022: 2149-2156.
>
> [5]Mormann F, Andrzejak RG. Seizure prediction: making mileage on the long and winding road. Brain. 2016 Jun;139(Pt 6):1625-7.
>
> [6]Truong N D, Nguyen A D, Kuhlmann L, et al. Integer convolutional neural network for seizure detection[J]. IEEE Journal on Emerging and Selected Topics in Circuits and Systems, 2018, 8(4): 849-857.

---

> > ### Comment · Reviewer_oXnd · 2023-08-21
> >
> > Thank you for addressing my comments. The references considered in the revised version are not much relevant and most of them are old. It is suggested to check the literature review and consider some recent and relevant related studies (e.g., published work on related problem).

---

> > > ### Author Response · Authors · 2023-08-21
> > > **Response to the new comment**
> > >
> > > Thanks for the suggestions. We update our literature review with some recent and relevant related studies as follows:
> > >
> > > **SEEG-based seizure detection.** SEEG is an emerging method applied in seizure detection, which can localize the SOZ more precisely than those noninvasive recording methods. However, due to the low-quality, large-amount, high-dimensionality characteristics of SEEG data, it is still challenging to develop an automatic approach in SEEG-based seizure detection. Ganti et al. [1] improve seizure detection by temporal Generative Adversarial Networks (TGAN). Chen et al. [2] adopt a graph structure to detect epileptic wave. Xiao et al. [3] propose an SOZ localization method via analyzing the long-term SEEG monitoring for preoperative planning of epilepsy surgery. Although researchers have explored some possible approaches for SEEG-based seizure detection, almost all of these works focus on a patient-specific setting, none of which can be applied in actual clinical scenarios.
> > >
> > > **Domain generalization on brain signal.** Our goal is to predict epileptic seizures of SEEG from unseen patients, which can be abstracted as a domain generalization (DG) problem on time series. Conceptually, DG deals with a challenging setting where one or several different but related domain(s) are given, and the goal is to learn a model that can generalize to an unseen test domain. With the development of DG researches in the fields of computer vision and natural language processing, related problems on brain signal also attract many research interests. Yang et al. [4] develop a new domain generalization method ManyDG, that can scale to such many-domain problems for seizure detection task on EEG. Ayodele et al. [5] use transfer component analysis and LSTM to detect epilepsy on EEG data. Jeon et al. [6] propose a mutual information-driven method to conduct subject-invariant and class-relevant deep representation learning of EEG. For these current DG works on brain signal, most of them are conducted on EEG data rather than more informative SEEG. Although Wang et al. [7] study SEEG-based seizure detection on the patient-independent setting, they conduct experiments on datasets which are not only much smaller in size than practical records. The datasets are also manually denoised and sampled to a balanced positive-negative sample ratio which brings about a huge data bias from the real clinical data, indicating that their work is still far from clinical requirements.
> > >
> > > Thanks again for your valuable suggestions which greatly improve our work.
> > >
> > > References:
> > >
> > > [1] Ganti B, Chaitanya G, Balamurugan RS, Nagaraj N, Balasubramanian K, Pati S. Time-Series Generative Adversarial Network Approach of Deep Learning Improves Seizure Detection From the Human Thalamic SEEG. Front Neurol. 2022
> > >
> > > [2]Chen J, Yang Y, Yu T, et al. Brainnet: Epileptic wave detection from seeg with hierarchical graph diffusion learning[C]//Proceedings of the 28th ACM SIGKDD Conference on Knowledge Discovery and Data Mining. 2022
> > >
> > > [3]Linxia Xiao, Caizi Li, Yanjiang Wang, Junxi Chen, Weixin Si, Chen Yao, Xifeng Li, Chuanzhi Duan, and Pheng-Ann Heng. Automatic localization of seizure onset zone from high-frequency seeg signals: A preliminary study. IEEE Journal of Translational Engineering in Health and Medicine, 2021
> > >
> > > [4] Yang, Chaoqi, M. Brandon Westover, and Jimeng Sun. "Manydg: Many-domain generalization for healthcare applications." *arXiv preprint arXiv:2301.08834*, 2023.
> > >
> > > [5] Kayode Peter Ayodele, Wisdom O Ikezogwo, Morenikeji A Komolafe, and Philip Ogunbona. Supervised domain generalization for integration of disparate scalp eeg datasets for automatic epileptic seizure detection. Computers in Biology and Medicine, 120:103757, 2020.
> > >
> > > [6] Eunjin Jeon, Wonjun Ko, Jee Seok Yoon, and Heung-Il Suk. Mutual information-driven subject invariant and class-relevant deep representation learning in bci. IEEE Transactions on Neural Networks and Learning Systems, 2021.
> > >
> > > [7] Yiping Wang, Yanfeng Yang, Gongpeng Cao, Jinjie Guo, Penghu Wei, Tao Feng, Yang Dai,  Jinguo Huang, Guixia Kang, and Guoguang Zhao. Seeg-net: An explainable and deep learning based cross-subject pathological activity detection method for drug-resistant epilepsy. Computers in Biology and Medicine, page 105703, 2022.

---

### Author Rebuttal · Authors · 2023-08-09

### Global response

We thank the reviewers for their close read of this manuscript and their insightful comments.

In response to reviewers' comments, we additionally performed 2 sets of experiments in the ablation study and 7 sets of experiments in the case study. Several important suggestions were made and we have considered each carefully and revised accordingly. Please find our replies to the reviewers below for detailed responses to the reviewers’ comments. We hope these updates address all key concerns and clarify the significance of our work.

**Note**

* In the global response, we **upload a pdf file**, which includes our additional experiments for the case study and two examples of SEEG signals with detection results.

* We conducted 2 additional experiments in ablation study on the reconstrcution loss and the self-attention. The results are shown in Table 1.

  * PPi-reconstruction: Pre-training w/o reconstrcution loss.
  * PPi-self attention: Aggregate the representations from time and frequency domains by a simple mean pooling w/o self-attention layer.

  Table 1: The ablation study on the reconstrcution loss and the self-attention.

  |                    | MAYO             |                  |                  |                  | FNUSA            |                  |                  |                  | Clinical         |                  |                  |                  |
  | ------------------ | ---------------- | ---------------- | ---------------- | ---------------- | ---------------- | ---------------- | ---------------- | ---------------- | ---------------- | ---------------- | ---------------- | ---------------- |
  |                    | Pre.             | Rec.             | F1               | F2               | Pre.             | Rec.             | F1               | F2               | Pre.             | Rec.             | F1               | F2               |
  | PPi-reconstruction | 44.01+/-11.56    | 50.29+/-10.39    | 34.15+/-8.26     | 36.89+/-8.75     | 60.78+/-11.33    | 62.31+/-9.79     | 59.48+/-7.9      | 60.02+/-6.86     | 22.04+/-8.38     | 44.84+/-10.93    | 21.23+/-6.41     | 25.01+/-7.96     |
  | PPi-self attention | 48.82+/-5.10     | 60.2+/-3.07      | 51.15+/-4.64     | 55.41+/-4.43     | 65.14+/-6.01     | 66.91+/-2.95     | 62.17+/-3.93     | 63.98+/-3.42     | 28.67+/-5.61     | 46.98+/-5.46     | 29.56+/-3.59     | 32.59+/-2.68     |
  | PPi                | **49.85+/-6.93** | **69.67+/-2.82** | **54.35+/-4.72** | **61.07+/-4.69** | **71.73+/-4.06** | **70.81+/-2.14** | **70.61+/-2.82** | **70.55+/-2.28** | **29.76+/-5.45** | **47.59+/-5.16** | **30.92+/-3.45** | **35.51+/-2.35** |

  We will add these two sets of experimental results to the original ablation study in our manuscript.

Thanks again for the thoughtful commentary. We have put in considerable effort to improve our manuscript, and we sincerely hope you will find our responses informative and helpful.

---

### Decision · Program_Chairs · 2023-09-21

**Decision:**

Accept (poster)

**Comment:**

The 6 reviewers here have expressed support on this work and did endorse it for publication. Given the thorough reviews and later discussions I support the publication of this work at NeurIPS 2023.